# ATTENTION IS ALL YOU NEED
# FOR KV CACHE IN DIFFUSION LLMs

**Quan Nguyen-Tri**[1,2*]**, Mukul Ranjan**[1*]**, Zhiqiang Shen**[1†]
[1]VILA Lab, MBZUAI, Abu Dhabi, UAE   [2]FPT Software AI Center, Vietnam
{quan.tri,mukul.ranjan,zhiqiang.shen}@mbzuai.ac.ae

Project page: https://vila-lab.github.io/elastic-cache-webpage/

## ABSTRACT

This work studies how to adaptively recompute key–value (KV) caches for diffusion large language models (DLMs) to maximize prediction accuracy while minimizing decoding latency. Prior methods' decoders recompute QKV for all tokens at every denoising step and layer, despite KV states changing little across most steps, especially in shallow layers, leading to substantial redundancy. We make three observations: (1) distant **MASK** tokens primarily act as a length-bias and can be cached block-wise beyond the active prediction window; (2) KV dynamics increase with depth, suggesting that selective refresh starting from deeper layers is sufficient; and (3) the most-attended token exhibits the smallest KV drift, providing a conservative lower bound on cache change for other tokens. Building on these, we propose **Elastic-Cache**, a training-free, architecture-agnostic strategy that jointly decides *when* to refresh (via an attention-aware drift test on the most-attended token) and *where* to refresh (via a depth-aware schedule that recomputes from a chosen layer onward while reusing shallow-layer caches and off-window MASK caches). Unlike fixed-period schemes, Elastic-Cache performs adaptive, layer-aware cache updates for diffusion LLMs, reducing redundant computation and accelerating decoding with negligible loss in generation quality. Experiments on LLaDA-Instruct, LLaDA-1.5, and LLaDA-V across mathematical reasoning and code generation tasks demonstrate consistent speedups: $8.7\times$ on GSM8K (256 tokens), and $45.1\times$ on longer sequences, while consistently maintaining higher accuracy than the baseline. Our method achieves significantly higher throughput ($6.8\times$ on GSM8K) than existing confidence-based approaches while preserving generation quality, enabling practical deployment of diffusion LLMs.

## 1 INTRODUCTION

Diffusion large language models (DLMs) (Li et al., 2025) have recently emerged as a compelling alternative to autoregressive Transformers (Radford et al., 2018; Achiam et al., 2023), yet their iterative denoising procedure makes inference particularly compute-intensive. In standard implementations, each decoding step recomputes queries, keys, and values (QKV) for every token at every layer, even though the underlying key–value (KV) states change only marginally across most steps. This all-tokens, all-layers recomputation incurs substantial latency and memory traffic, ultimately limiting practical deployment. Our goal in this study is to determine *how and when* to adaptively recompute the KV cache during decoding so as to maximize prediction quality while minimizing wall-clock latency.

A defining property of diffusion LLM decoding is the progressive unmasking of tokens under a length- and structure-aware attention pattern. This induces heterogeneous KV dynamics: shallow layers tend to stabilize quickly as they encode local lexical structure, whereas deeper layers continue to adjust global, semantic dependencies. We formalize this with a notion of KV drift: the step-to-step change in cached keys and values, and observe two consistent trends: (i) drift is small for most

---

*Equal contribution. †Corresponding author.

steps, and (ii) drift grows with layer depth. These trends suggest that indiscriminate recomputation is wasteful, and that targeted refreshes could preserve accuracy while slashing cost.

Prior acceleration methods for diffusion (and related) decoders typically refresh the KV cache on a fixed schedule, e.g., every $k$ iterations without regard to instance difficulty, current attention patterns, or layerwise variability. Such fixed-period policies leave performance on the table: they recompute when nothing has changed and miss updates precisely when rapid semantic revisions occur. Moreover, by treating all layers uniformly, they over-service shallow layers whose representations have already converged, while under-servicing deeper layers where changes matter most. This motivates an adaptive, attention-aware alternative.

Our approach is built on three empirical observations. First, distant MASK tokens exert negligible influence on unmasking the current token and behave primarily as a length-bias prior; thus, their KV can be block-cached outside the active prediction window to avoid redundant work. Second, KV drift increases with depth, so refreshes should start at an automatically learned boundary layer $\ell^\star$ (determined by attention threshold and adapted to each input decoding step) and apply only to deeper layers, reusing shallow-layer caches. Third, the most-attended token at a step typically exhibits the smallest drift, providing a conservative lower bound on KV changes across the context. Monitoring this drift yields a reliable, low-overhead trigger for deciding whether a global refresh is warranted.

Based on these ideas, we propose **Elastic-Cache**, a training-free, architecture-agnostic strategy that couples *Attention-Aware KV Cache Update* with *Layer-Aware KV Cache Update*. The attention-aware module computes a lightweight drift statistic on the most-attended token; if the statistic exceeds a threshold, a refresh is triggered, otherwise cached KVs are reused. The layer-aware module then refreshes only layers $\ell \geq \ell^\star$, while shallow layers retain their caches, and off-window MASK tokens remain block-cached. Together, these mechanisms align recomputation with *where and when* the model's beliefs actually change, minimizing unnecessary QKV work.

In contrast to fixed-period baselines, our **Elastic-Cache** adapts to the input, step, and layer granularity together. It reduces compute by skipping recomputation during stable phases, focuses effort on deeper layers during semantic revisions, and leverages block-wise caching for distant MASK tokens. Conceptually, the method reframes KV management as an **attention-guided control problem**: attention estimates *which* tokens matter; drift detects *how much* the state has changed; and the layer boundary $\ell^\star$ encodes *where* updates pay off. This yields a practical pathway to low-latency diffusion LLM decoding without modifying training or the base architecture.

Our contributions of this work:

- We diagnose redundancy in diffusion LLM decoding and introduce KV drift as a principled signal for adaptive cache management.
- We propose **Elastic-Cache**, the first (to our best knowledge) adaptive, layer-aware KV refresh policy for diffusion LLMs that jointly decides *when* to recompute (attention-aware drift test) and *where* to recompute (depth-selective updates).
- We develop *block-wise MASK caching* to eliminate needless updates outside the prediction window. We provide comprehensive empirical experiments and ablations showing that our **Elastic-Cache** preserves generation quality while substantially reducing decoding latency across tasks and model scales.

## 2 PRELIMINARY

### 2.1 MASKED DIFFUSION MODELS

Masked Diffusion Models (MDMs), absorbing-state discrete diffusion, build on D3PM (Austin et al., 2021a) and its continuous-time variant (Campbell et al., 2022), replacing tokens with a special MASK along a forward process (Sahoo et al., 2024; Shi et al., 2024) at timestep $t$:

$$q_{t|0}(\boldsymbol{x}_t|\boldsymbol{x}_0) = \prod_{i=1}^{L} q_{t|0}(x_t^i|x_0^i) = \prod_{i=1}^{L} \text{Cat}(x_t^i; (1-t)\delta_{x_0^i} + t\delta_{\text{MASK}}) \tag{1}$$

where $t \in [0,1]$ controls interpolation between the original data $\boldsymbol{x}_0$ (at $t = 0$) and a fully masked sequence (at $t = 1$), $\text{Cat}(\cdot)$ denotes the categorical distribution. A parametric model $p_{\boldsymbol{\theta}}$ learns the reverse denoising; generation starts from all MASK and iteratively unmasks by sampling $p_{\boldsymbol{\theta}}(x_0^i|\boldsymbol{x}t)$.

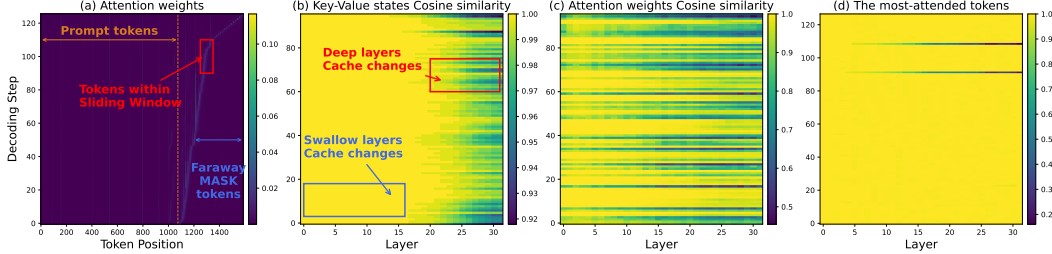

Figure 1: Visualization of our motivation. (a) MASK tokens located near each other receive high attention, while those situated far apart have minimal influence. (b) Over time, the representations in the KV states of cached tokens evolve, with deeper layers experiencing more substantial changes. (c) The changes in attention weights of most-attended tokens exhibit similar patterns to the changes in KV states of all cached tokens. (d) KV states of the most-attended tokens have the least changes.

Recent theory (MDLM (Shi et al., 2024; Sahoo et al., 2024), RADD (Ou et al., 2024)) simplifies training from a variational bound to a reweighted cross-entropy over masked positions:

$$\mathcal{L}_{\text{MDM}} = \int_0^1 \frac{1}{t} \mathbb{E}_{q_{t|0}(\boldsymbol{x}_t|\boldsymbol{x}_0)} \left[ \sum_{i:x_t^i=\text{MASK}} -\log p_{\boldsymbol{\theta}}(x_0^i|\boldsymbol{x}_t) \right] \mathrm{d}t \tag{2}$$

This formulation scales to LLMs as diffusion language models (DLMs), with LLaDA (Nie et al., 2025b) and Dream-7B (Ye et al., 2025) matching autoregressive performance while enabling parallel decoding and flexible infilling.

## 2.2 KEY-VALUE CACHE IN TRANSFORMERS

Transformer-based language models achieve computational efficiency during autoregressive generation through Key-Value (KV) caching (Pope et al., 2023). In causal attention, each layer projects the current hidden state $\mathbf{H}^t$ into query, key, and value representations using learned projection matrices $\mathbf{W_Q}, \mathbf{W_K}, \mathbf{W_V}$. At decoding step $t$, the attention computation for the current token follows:

$$\mathbf{A}_{[t]}^t = \text{softmax} \left( \frac{\mathbf{Q}_{[t]}^t (\mathbf{K}_{[1:t]}^t)^\top}{\sqrt{d_k}} \right) \mathbf{V}_{[1:t]}^t, \quad \text{KV cache:} \begin{cases} \mathbf{K}_{[1:t]}^t = \text{concat}(\mathbf{K}_{[1:t-1]}^{t-1}, \mathbf{K}_{[t]}^t), \\ \mathbf{V}_{[1:t]}^t = \text{concat}(\mathbf{V}_{[1:t-1]}^{t-1}, \mathbf{V}_{[t]}^t) \end{cases} . \tag{3}$$

To avoid redundant computation, previous key-value pairs are cached and reused. This caching strategy is effective because in causal attention, previously computed key-value pairs remain invariant throughout decoding ($\mathbf{K}_{[1:t-1]}^{t-1} = \mathbf{K}_{[1:t-1]}^t$), enabling efficient reuse without affecting model output.

**KV-Cache in Bidirectional Attention.** However, in diffusion models, bidirectional attention allows all positions to attend to each other, invalidating the invariance assumption of traditional KV-cache. As dKV-Cache (Ma et al., 2025) observes, token representations evolve across denoising steps, making cached keys/values stale. This dynamic behavior necessitates rethinking caching strategies for diffusion language models.

## 3 METHODOLOGY

### 3.1 OUR FRAMEWORK OVERVIEW AND MOTIVATION

Diffusion LLMs differ from autoregressive decoders in that their key–value (KV) states evolve across denoising steps due to bidirectional dependencies. Our objective is to adaptively decide *when* and *where* to recompute the KV cache to preserve accuracy while minimizing latency. Baseline decoders recompute QKV for all tokens and layers at every step, despite *negligible* KV changes for most steps and especially in *shallow* layers (Fig. 1b); deeper layers exhibit larger drift. Rather than fixed-period refreshes (Wu et al., 2025; Ma et al., 2025; Liu et al., 2025), we propose **Elastic-Cache**, the first (to our knowledge) *adaptive, layer-aware* KV update policy for diffusion LLMs that jointly optimizes timing and location of recomputation.

Our design is driven by three observations. (1) Distant MASK tokens mainly act as a length prior and exert minimal influence on the current unmasking, we therefore block-cache their KV beyond the

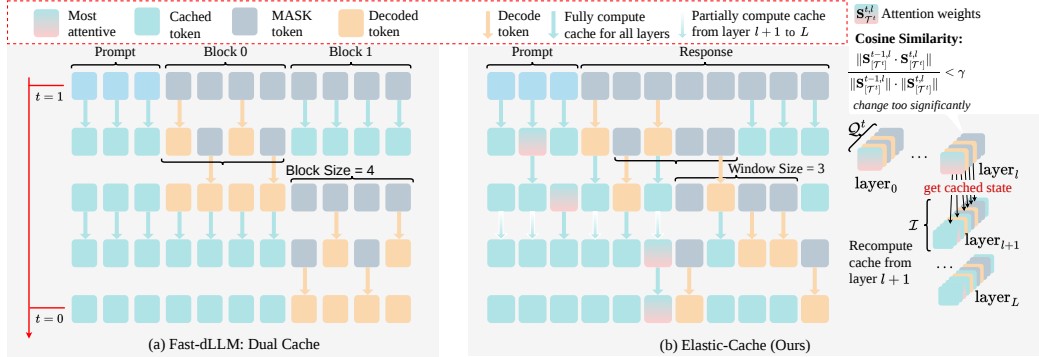

Figure 2: Illustration of the Key-Value cache method for diffusion LLMs. (a) The fast-dLLM (Wu et al., 2025) block-wise decoding method caches the Key-Value of all tokens outside the current block at each step. The KV cache is updated after completing a block of decoding. (b) Our proposed method, Elastic-Cache, caches the key-value of tokens outside a sliding window that flexibly moves through the sentence from left to right at each iteration. When the attention weights corresponding to the most-attended tokens (one for each layer) change significantly at a layer $l$, we start recomputing the KV cache from layer $l + 1$ to the last layer.

active prediction window (Fig. 1a). (2) KV drift grows with depth, refresh should start at a boundary layer and apply only to deeper layers (Fig. 1b). (3) The most-attended tokens typically shows the smallest KV change (Fig. 1d), giving a conservative lower bound for others, we use its drift as a lightweight trigger for refresh (Fig. 1c). Fig. 2 summarizes the pipeline. To the end, we proposed Elastic-Cache, a flexible method for key-value caching in diffusion large language models. Fig. 2 provides a visual representation of the overall pipeline of our proposed method.

## 3.2 SLIDING WINDOW DECODING AND KV CACHING

Formally, let $\mathcal{I} = \{1, 2, \ldots, N\}$ represent all positions. At decoding step $t$, let $\mathcal{D}^t$ denote newly decoded positions and $\mathcal{M}^t$ denote remaining masked positions, where $\mathcal{M}^{t-1} = \mathcal{M}^t \cup \mathcal{D}^t$. Denotes $\mathcal{D}^{<t} = \bigcup_i \{\mathcal{D}^i\}_{i=1}^t$ as the set of all decoded tokens up to time step $t$. Initially, at $t = 0$ we compute the attention for each layer $l$:

$$\mathbf{A}_{[\mathcal{I}]}^{0,l} = \text{softmax}\left(\frac{\mathbf{Q}_{[\mathcal{I}]}^{0,l}(\mathbf{K}_{[\mathcal{I}]}^{0,l})^\top}{\sqrt{d_k}}\right)\mathbf{V}_{[\mathcal{I}]}^{0,l}, \qquad \text{initialize KV cache:} \begin{cases} \tilde{\mathbf{K}}_{[\mathcal{I}]}^{0,l} = \mathbf{K}_{[\mathcal{I}]}^{0,l} \\ \tilde{\mathbf{V}}_{[\mathcal{I}]}^{0,l} = \mathbf{V}_{[\mathcal{I}]}^{0,l} \end{cases}. \quad (4)$$

For each subsequence iteration $t$ ranging from 1 to $T$, The model perform prediction for newly decoded position $\mathcal{D}^t$ and the remaining masked position $\mathcal{M}^t$. To enhance efficiency, we only perform predictions for masked positions that are closest to the left and form a sliding window of size $\beta$, denoted as $\mathcal{M}_\beta^t = \mathcal{M}_{[1:\beta]}^t$. We also have that $\mathcal{M}_\beta^{t-1} = \mathcal{M}_\beta^t \cup \mathcal{D}^t$. We observe that masked tokens within the sliding window attend closely to one another, while those outside receive little attention. This allows safe reuse of cached KV for out-of-window MASKs without affecting current predictions. At step $t$, attention is computed only for tokens in the sliding window $\mathcal{M}_\beta^{t-1}$:

$$\mathbf{A}_{[\mathcal{M}_\beta^{t-1}]}^{t,l} = \text{softmax}\left(\frac{\mathbf{Q}_{[\mathcal{M}_\beta^{t-1}]}^{t,l}(\tilde{\mathbf{K}}_{[\mathcal{I}]}^{t,l})^\top}{\sqrt{d_k}}\right)\tilde{\mathbf{V}}_{[\mathcal{I}]}^{t,l}, \quad \text{update cache:} \begin{cases} \tilde{\mathbf{K}}_{[\mathcal{M}_\beta^{t-1}]}^{t,l} = \mathbf{K}_{[\mathcal{M}_\beta^{t-1}]}^{t,l} \\ \tilde{\mathbf{V}}_{[\mathcal{M}_\beta^{t-1}]}^{t,l} = \mathbf{V}_{[\mathcal{M}_\beta^{t-1}]}^{t,l} \end{cases}. \quad (5)$$

While sliding window decoding shares similarities with Fast-dLLM's block-wise KV caching (Wu et al., 2025) (see Fig. 2), it offers key improvements. By predicting nearby tokens together, it reduces cache loss for distant MASK tokens. In contrast, block-wise decoding may miss MASK tokens near block ends, resulting in less efficient predictions due to overly aggressive context caching.

## 3.3 ATTENTION-AWARE KV CACHE UPDATE

The most important novelty of our proposed method is to automatically determine whether to update the KV cache to preserve accuracy while minimizing latency. Our method leverages the awareness of the model's attention weights to identify when the KV cache undergoes significant changes. At

---

**Algorithm 1** The Elastic-Cache Algorithm

---

1: **Require:** Prompt $\mathbf{x}_{\text{prompt}}$, Sliding window size $\beta$, Update threshold $\gamma$, Generation length $N$.
2: **Initialize:** $\mathbf{x}^0 \leftarrow \{\mathbf{x}_{\text{prompt}}; \texttt{[MASK]}, \ldots, \texttt{[MASK]}\}; \quad P \leftarrow \texttt{length}(\mathbf{x}_{\text{prompt}});$
3: $t \leftarrow 1; \quad \mathcal{D}^1 \leftarrow \{1, \ldots, P\}; \quad \mathcal{M}^1 \leftarrow \{P+1, \ldots, P+N\}; \quad \mathcal{T}^0 \leftarrow \emptyset;$
4: **while** $\mathcal{M}^t \neq \emptyset$ **do**
5: $\quad \mathcal{M}_\beta^t \leftarrow \mathcal{M}_{[:\beta]}^t; \quad \mathcal{Q}^t \leftarrow \mathcal{T}^{t-1} \cup \mathcal{M}_\beta^{t-1}; \quad \mathbf{H}_{[\mathcal{Q}^t]}^{t,0} \leftarrow \texttt{Embedding}(\mathbf{x}_{[\mathcal{Q}^t]}^t); \quad l^* \leftarrow \infty$
6: $\quad$ **for** $l = 1, \ldots, L$ **do**
7: $\quad\quad$ **if** $l > l^*$ **then** *// Cache Update*
8: $\quad\quad\quad \tilde{\mathbf{H}}_{[\mathcal{I}]}^{t,l}, \tilde{\mathbf{K}}_{[\mathcal{I}]}^{t,l}, \tilde{\mathbf{V}}_{[\mathcal{I}]}^{t,l} \leftarrow \texttt{cache}(\mathcal{I}); \quad \mathbf{Q}_{[\mathcal{I}]}^{t,l}, \mathbf{K}_{[\mathcal{I}]}^{t,l}, \mathbf{V}_{[\mathcal{I}]}^{t,l} = \texttt{FFN}(\mathbf{H}_{[\mathcal{I}]}^{t,l})$
9: $\quad\quad\quad \mathbf{H}_{[\mathcal{I}]}^{t,l+1}, \mathbf{S}_{[\mathcal{T}^{t-1}]}^{t,l} \leftarrow \texttt{MHA}(\mathbf{Q}_{[\mathcal{I}]}^{t,l}, \mathbf{K}_{[\mathcal{I}]}^{t,l}, \mathbf{V}_{[\mathcal{I}]}^{t,l})$
10: $\quad\quad$ **else** *// Cache Reuse*
11: $\quad\quad\quad \tilde{\mathbf{H}}_{[\mathcal{Q}^t]}^{t,l}, \tilde{\mathbf{K}}_{[\mathcal{Q}^t]}^{t,l}, \tilde{\mathbf{V}}_{[\mathcal{Q}^t]}^{t,l} \leftarrow \texttt{cache}(\mathcal{Q}^t); \quad \mathbf{Q}_{[\mathcal{Q}^t]}^{t,l}, \mathbf{K}_{[\mathcal{Q}^t]}^{t,l}, \mathbf{V}_{[\mathcal{Q}^t]}^{t,l} = \texttt{FFN}(\mathbf{H}_{[\mathcal{Q}^t]}^{t,l})$
12: $\quad\quad\quad \mathbf{H}_{[\mathcal{Q}^t]}^{t,l+1}, \mathbf{S}_{[\mathcal{T}^{t-1}]}^{t,l} \leftarrow \texttt{MHA}(\mathbf{Q}_{[\mathcal{Q}^t]}^{t,l}, \tilde{\mathbf{K}}_{[\mathcal{I}]}^{t,l}, \tilde{\mathbf{V}}_{[\mathcal{I}]}^{t,l})$
13: $\quad\quad\quad \sigma^{t,l} \leftarrow \texttt{cosine\_similarity}(\mathbf{S}_{[\mathcal{T}^{t-1}]}^{t-1,l}, \mathbf{S}_{[\mathcal{T}^{t-1}]}^{t,l})$
14: $\quad\quad\quad$ **if** $\sigma^{t,k} < \gamma$ **then** *// Start update cache from layer $l+1$*
15: $\quad\quad\quad\quad l^* \leftarrow l; \quad \mathbf{H}_{[\mathcal{I}]}^{t,l+1} \leftarrow \texttt{get\_cached\_state}(\mathbf{H}_{[\mathcal{Q}^t]}^{t,l+1})$
16: $\quad\quad\quad$ **end if**
17: $\quad\quad$ **end if**
18: $\quad\quad$ Get the most-attended token: $\mathcal{T}^{t,l} \leftarrow \arg\max_{k \in \mathcal{D}^{<t}} \sum_{q \in \mathcal{M}_\beta^t} \mathbf{S}_{[q,k]}^{t,l}$
19: $\quad$ **end for**
20: $\quad$ Decode new tokens: $\mathbf{x}^{t+1}, \mathcal{D}^{t+1} \leftarrow \texttt{decode}(\mathbf{x}^t, \mathcal{M}_\beta^t)$
21: $\quad$ Update state: $\mathcal{M}^{t+1} \leftarrow \mathcal{M}^t \setminus \mathcal{D}^{t+1}; \quad \mathcal{T}^t = \bigcup_l \{\mathcal{T}^{t,l}\}_{l=1}^L \quad t \leftarrow t+1$ *// State Update*
22: **end while**
23: **return** $\mathbf{x}^{t-1}$

---

time step $t$ and layer $l$, we determine the token that receives the most frequent attention from other tokens based on the attention weights corresponding to the current model's prediction for $\mathcal{M}_\beta^t$.

$$\mathcal{T}^{t,l} = \arg\max_{k \in \mathcal{D}^{<t}} \sum_{q \in \mathcal{M}_\beta^t} \mathbf{S}_{[q,k]}^{t,l}, \qquad \text{where: } \mathbf{S}_{[\mathcal{M}_\beta^t]}^{t,l} = \text{softmax}\left(\frac{\mathbf{Q}_{[\mathcal{M}_\beta^t]}^{t,l}(\tilde{\mathbf{K}}_{[\mathcal{I}]}^{t,l})^\top}{\sqrt{d_k}}\right). \quad (6)$$

Here, we focus solely on the most-attended token among the current decoded tokens $\mathcal{D}^{<t}$. This is because the remaining MASK tokens either fall within the sliding window of predictions or have negligible influence on the unmasking tokens (Fig. 1a). We obtain one most-attended token per layer and compile the set of most-attended tokens, denoted as $\mathcal{T}^t = \bigcup_l \{\mathcal{T}^{t,l}\}_{l=1}^L$. In practice, the most-attended token for a layer often overlaps with tokens from other layers, resulting in a relatively limited number of most-attended tokens being available at any given time.

$\mathcal{T}^t$, besides being the tokens that have the most influence on the predictions' outcome, also signify the least changes among the cached decoded tokens (Fig. 1d). Therefore, we use $\mathcal{T}^t$ as a lightweight trigger for our cache update mechanism. Without updating all cached tokens, we only frequently update the most-attended tokens $\mathcal{T}^t$ to measure the degree of changes for all other cached tokens. Ideally, since $\mathcal{T}^t$ have the least change among the decoded tokens, we expect that when $\mathcal{T}^t$ change significantly, the rest of the decoded tokens will also change significantly. Therefore, we add $\mathcal{T}^{t-1}$ to the sliding window at step $t$: $\mathcal{T}^{t-1} \cup \mathcal{M}_\beta^{t-1}$. We then measure the changes in attention weights of $\mathcal{T}^t$ between the current and previous steps, $t$ and $t-1$, using cosine similarity.

$$l^* = \begin{cases} l & \text{if: } \sigma^{t,l} < \gamma \\ \infty & \text{othewise} \end{cases}, \quad \text{Cosine Similarity: } \quad \sigma^{t,l} = \frac{\|\mathbf{S}_{[\mathcal{T}^{t-1}]}^{t-1,l} \cdot \mathbf{S}_{[\mathcal{T}^{t-1}]}^{t,l}\|}{\|\mathbf{S}_{[\mathcal{T}^{t-1}]}^{t-1,l}\| \cdot \|\mathbf{S}_{[\mathcal{T}^{t-1}]}^{t,l}\|}. \quad (7)$$

The changes in attention $\mathbf{S}^{t,l}$ directly affect the output of the current layer or the input of the next layer $\mathbf{H}^{t,l+1}$. This implies that our cached values are diverging from the actual values, necessitating an update. When a layer $l^*$ observes significant changes in attention weights $\sigma^{t,l} < \gamma$, we initiate the update of the KV cache for the subsequent layers, starting from $l^* + 1$ and continuing until the last layer $L$. To achieve this, we initialize the hidden states of all cached tokens with the states $\tilde{\mathbf{H}}_{[\mathcal{I}]}^{t,l+1}$,

which have been saved and updated using patterns similar to $\tilde{\mathbf{K}}_{[\mathcal{I}]}^{t,l+1}$ and $\tilde{\mathbf{V}}_{[\mathcal{I}]}^{t,l+1}$.

$$\text{Update state: } \tilde{\mathbf{H}}_{[\mathcal{M}_\beta^{t-1}]}^{t,l+1} = \mathbf{H}_{[\mathcal{M}_\beta^{t-1}]}^{t,l+1}, \quad \text{Initialize: } \mathbf{Q}_{[\mathcal{I}]}^{t,l+1}, \mathbf{K}_{[\mathcal{I}]}^{t,l+1}, \mathbf{V}_{[\mathcal{I}]}^{t,l+1} = \text{linear}(\tilde{\mathbf{H}}_{[\mathcal{I}]}^{t,l+1}) \quad (8)$$

We then update and overwrite the KV cache using the same process as initially at $t = 0$, as described in Eq. 4. If none of the layers satisfy $\sigma^{t,l} < \gamma$, we continue to reuse our KV cache for future predictions.

We didn't directly compare the hidden state $\mathbf{H}^{t,l+1}$ and $\mathbf{H}^{t-1,l+1}$ because their changes depend on various network components. The error in measurement could be amplified by the divergence between the cached value and the actual value (including Key-Value states).

On the other hand, the changes in attention weights are closely linked to the source of the change in Key-Value states, which is the bidirectional attention mechanism in diffusion LLMs. Intuitively, the changes in attention weights become significant when new decoded tokens receive high attention and alter the attention output computed in the past when they were still masked. Consequently, the changes in attention weights exhibit very similar patterns to the changes in Key-Value states during decoding, as illustrated in Fig. 1b and Fig. 1c.

Our approach is formally grounded in **Theorem A.9** (Appendix), which proves that the most-attended token $\mathcal{T}^{t,\ell}$ has KV drift bounded by $\Delta_{\mathcal{T}^{t,\ell}}^{t,\ell} \leq \bar{\Delta}^{t,\ell} + O\left(\frac{\sqrt{d_k}}{R_\ell \sqrt{N}}\right)$, where the error term scales negligibly with typical transformer dimensions. This establishes that monitoring attention patterns of most-attended tokens provides a computationally efficient and theoretically sound proxy for overall KV drift.

We use the hyper-parameter $\gamma$ to set the trigger for automatic cache updates. As shown in Fig. 1c, the attention weights' cosine similarity landscapes influence this. A higher $\gamma$ results in more frequent and extensive cache updates across multiple layers, while a lower $\gamma$ triggers updates less frequently. This flexibility allows us to effectively manage the trade-off between accuracy and latency. Our overall algorithm is described in Algorithm 1.

## 4 EXPERIMENTS

### 4.1 EXPERIMENTAL SETUP

**Implementation Details.** All our runs use a single NVIDIA A100 80GB. We evaluate **Elastic-Cache** on LLaDA-Instruct (Nie et al., 2025a), LLaDA-1.5 (Zhu et al., 2025), and multimodal LLaDA-V (You et al., 2025) across MBPP (Austin et al., 2021b), HumanEval (Chen et al., 2021), MATH (Hendrycks et al., 2021), GSM8K (Cobbe et al., 2021), MathVista (Lu et al., 2023), and MathVerse (Zhang et al., 2024b). Default hyperparameters: attention threshold $\gamma$=0.9, parallel-decoding confidence $\epsilon$=0.9, cache block size 32. For fair comparison, we re-run LLaDA (Nie et al., 2025a) and Fast-dLLM (Wu et al., 2025) under the same hardware/software. **Evaluation Framework and Metrics.** We use `lm-eval-harness` (Gao et al., 2024). Throughput is tokens/sec averaged until emitting, matching Fast-dLLM's protocol (Wu et al., 2025). Accuracy metrics: GSM8K: 5-shot `flexible_extract` (Cobbe et al., 2021); MATH: 4-shot `math_verify` (`minerva_math`) (Hendrycks et al., 2021); HumanEval—0-shot with the Fast-dLLM post-processing (Chen et al., 2021; Wu et al., 2025); MBPP—3-shot `pass@1` (Austin et al., 2021b). For LLaDA-V, we adopt the official pipeline with `lmms-eval` (Zhang et al., 2024a; Li et al., 2024): MathVista: `gpt_eval_score` (Lu et al., 2023); MathVerse: `gpt_eval_score` on `mathverse_testmini_vision_dominant` (Zhang et al., 2024b).

**Confidence-Aware Decoding.** We employ confidence-aware decoding strategies from Fast-dLLM (Wu et al., 2025), which select only tokens with confidence scores exceeding a specified threshold ($\epsilon$), instead of unmasking a fixed number of tokens per step, as in the baseline Diffusion LLM. This straightforward yet effective approach accelerates Diffusion LLM inference by enabling more tokens to be predicted concurrently at each iteration, contingent upon the model's performance. Consequently, we concentrate on comparing the acceleration achieved by the KV caching method under the same decoding strategies.

### 4.2 PERFORMANCE AND EFFICIENCY EVALUATION

Across Tables 1, 3, and 5, our proposed **Elastic-Cache** delivers substantial throughput gains for diffusion LLMs with minimal accuracy loss. By adaptively updating the cache only when neces-

Table 1: Comprehensive benchmark results on the LLaDA-Instruct suite. Each cell shows accuracy (top) and decoding throughput in tokens/sec with relative speedup to the LLaDA baseline (bottom, blue: t/s / orange: speedup). Highlighted cells denote the highest throughput and speedup per configuration. The highest accuracy is bolded.

| Benchmark | Gen Length | LLaDA | Confident-Aware Decoding | | |
| | | | LLaDA | Fast-dLLM | Elastic-Cache |
|---|---|---|---|---|---|
| GSM8K (5-shot) | 256 | 78.01 
 7.3 (1.0×) | 78.62 
 22.8 (3.1×) | 77.94 
 53.7 (7.7×) | **78.24** 
 58.0 (8.2×) |
| | 512 | 77.10 
 3.6 (1.0×) | 77.33 
 18.6 (5.2×) | 74.83 
 44.0 (12.3×) | **77.71** 
 90.1 (25.2×) |
| MATH (4-shot) | 256 | 33.58 
 9.5 (1.0×) | 33.28 
 25.8 (2.7×) | 32.50 
 49.0 (5.1×) | **33.14** 
 48.7 (5.1×) |
| | 512 | 37.20 
 7.1 (1.0×) | 36.82 
 24.0 (3.4×) | 35.70 
 52.8 (7.4×) | **36.60** 
 59.3 (7.9×) |
| HumanEval (0-shot) | 256 | 40.85 
 33.3 (1.0×) | 42.07 
 102.1 (3.1×) | 37.20 
 99.8 (3.0×) | **40.24** 
 160.5 (4.8×) |
| | 512 | 43.90 
 17.7 (1.0×) | 43.29 
 51.6 (2.9×) | 45.73 
 76.1 (4.3×) | **46.34** 
 100.7 (5.0×) |
| MBPP (3-shot) | 256 | 29.80 
 6.5 (1.0×) | 30.00 
 23.4 (3.6×) | 25.40 
 45.1 (7.0×) | **32.2** 
 46.9 (7.3×) |
| | 512 | 15.0 
 4.7 (1.0×) | 15.0 
 20.8 (4.4×) | 13.6 
 44.7 (9.5×) | **15.6** 
 63.0 (13.4×) |

Table 2: Comparison with additional KV caching methods on GSM8K (5-shot, 512 tokens) using LLaDA-1.5. Each cell shows accuracy (top) and throughput in t/s with relative speedup (bottom, blue: t/s; orange: speedup).

| LLaDA-1.5 | dKV-Cache | dLLM-Cache | DeepCache (N=10) | DeepCache (N=20) | Elastic-Cache |
|---|---|---|---|---|---|
| 81.35 
 2.6 (1.0×) | 67.02 
 14.82 (5.7×) | 80.97 
 16.84 (6.5×) | 83.1 
 58.4 (22.5×) | 81.4 
 60.9 (23.4×) | **83.7** 
 139.4 (53.6×) |

sary, it achieves a speedup of up to 45.1× over the standard baseline. While maintaining accuracy within 1∼2% on MATH and HumanEval, it also achieves higher accuracy on GSM8K and MBPP. Compared to Fast-dLLM (Wu et al., 2025), Elastic-Cache consistently attains greater tokens/sec at better accuracy. Elastic-Cache also generalizes to Dream-7B (Table 4), achieving 21.4× and 5.5× speedups on GSM8K and HumanEval respectively.

As presented in Table 1, on LLaDA-Instruct, **Elastic-Cache** reaches 90.1 t/s on GSM8K (512 tokens; 25.2× over baseline) at 77.71% accuracy, surpassing Fast-dLLM's 44.0 t/s @ 74.83%. On LLaDA-1.5 (Table 3), our approach yields even greater gains, including 45.1× on GSM8K with 512 Gen Length, with an accuracy of 81.35% (baseline 81.35%). This observation indicates that Elastic-Cache performs better when the model's predictions are more accurate. The reason behind this could be the close relationship between our approach and attention scores. Intuitively, accurate predictions are associated with meaningful attention scores with fewer outliers, which makes our approach operate more smoothly.

We also observed that in most settings, Elastic-Cache provides higher throughput for longer generation lengths, whereas this is the opposite for Fast-dLLM (Wu et al., 2025), as it often experiences reduced throughput as the generation length increases. The advantages of our approach stem from the fixed-size sliding window and automatic cache update, which minimizes the dependency of throughput on the generation length.

In the multimodal setting (LLaDA-V; Table 5), **Elastic-Cache** raises MathVerse-256 throughput to 32.3 t/s from Fast-dLLM's 30.3 t/s while maintaining 29.19% accuracy, demonstrating robustness beyond text-only tasks. The significant improvement of Elastic-Cache over baselines across various settings suggests that our method is broadly applicable and has high scalability potential.

**Comparison with Additional KV Caching Methods.** To further validate the effectiveness of Elastic-Cache, we compare against dLLM-Cache (Ma et al., 2025) and DeepCache (Ma et al., 2024) on GSM8K (512 tokens) with LLaDA-1.5 in Table 2. DeepCache uses fixed-interval cache updates with intervals $N = 10$ and $N = 20$. Our method achieves 139.4 t/s at 83.7% acc, significantly outperforming dLLM-Cache (16.84 t/s, 80.97%) and DeepCache variants (58.4-60.9 t/s, 81.4-83.1%), demonstrating the advantages of adaptive, attention-aware cache over fixed-schedule approaches.

Table 3: Comprehensive benchmark results on the LLaDA-1.5 suite. Each cell shows accuracy (top) and decoding throughput in tokens/sec with relative speedup to the LLaDA baseline (bottom, blue: t/s; orange: speedup). Bold cells denote the highest throughput and speedup per configuration.

| Benchmark | Gen Length | LLaDA-1.5 | Confident-Aware Decoding | | |
| | | | LLaDA-1.5 | Fast-dLLM | Elastic-Cache |
|---|---|---|---|---|---|
| GSM8K (5-shot) | 256 | 80.36
6.7 (1.0×) | 80.44
22.5 (3.3×) | 80.59
51.2 (7.6×) | **81.50**
58.0 (8.7×) |
| | 512 | 81.35
2.6 (1.0×) | 81.88
17.2 (6.6×) | 80.82
36.8 (14.1×) | **81.35**
117.2 (45.1×) |
| MATH (4-shot) | 256 | 33.52
8.5 (1.0×) | 33.60
22.3 (2.6×) | 32.74
44.4 (5.2×) | **33.50**
51.0 (6.5×) |
| | 512 | 35.63
5.0 (1.0×) | 35.56
20.3 (4.0×) | 33.68
44.4 (8.8×) | **35.36**
74.8 (14.9×) |
| HuamnEval (0-shot) | 256 | 43.29
7.0 (1.0×) | 42.68
17.5 (2.5×) | 34.75
18.7 (2.7×) | **36.59**
20.9 (3.0×) |
| | 512 | 40.85
3.2 (1.0×) | 39.63
9.7 (3.1×) | 36.59
15.4 (4.8×) | **37.80**
16.8 (5.3×) |
| MBPP (3-shot) | 256 | 38.00
2.4 (1.0×) | 38.00
14.2 (5.8×) | 34.60
28.0 (11.6×) | **41.20**
32.7 (13.5×) |
| | 512 | 38.20
1.0 (1.0×) | 38.60
11.5 (11.5×) | 36.20
17.8 (17.8×) | **39.00**
32.8 (32.8×) |

Table 4: Comprehensive benchmark results on the "Dream-v0-Base-7B" suite. Each cell shows accuracy (top) and decoding throughput in tokens/sec with relative speedup to the Dream baseline (bottom, blue: t/s; orange: speedup).

| Benchmark | Gen Length | Dream | Confident-Aware Decoding | |
| | | | Fast-dLLM | Elastic-Cache |
|---|---|---|---|---|
| GSM8K (5-shot) | 512 | 76.0
7.9 (1.0×) | 74.1
45.9 (5.8×) | **75.6**
169.4 (21.4×) |
| HumanEval (0-shot) | 512 | 54.3
17.2 (1.0×) | 51.2
50.1 (2.9×) | **56.7**
95.2 (5.5×) |

Table 5: Performance and Speedup Comparison of LLaDA-V on MathVista and MathVerse. Each benchmark presents results from LLaDA-V (base) using Fast-dLLM, and our method.

| Length | MathVista | | | MathVerse | | |
| | Base Model (LLaDA-V) | Fast-dLLM | Elastic-Cache (Ours) | Base Model (LLaDA-V) | Fast-dLLM | Elastic-Cache (Ours) |
|---|---|---|---|---|---|---|
| 256 | 54.6 (2.3) | 55.9 (28.7) | 55.9 (29.7) | 30.1 (2.1) | 26.8 (30.3) | 29.2 (32.3) |
| 512 | 53.0 (1.9) | 54.1 (23.7) | 55.8 (24.1) | 26.9 (2.0) | 25.5 (28.1) | 29.2 (30.8) |

## 4.3 ABLATIONS

We ablate key choices: 1) Cache update threshold $\gamma$, 2) sliding window size $\beta$, and 3) prefill and generation length, to expose speed/accuracy trade-offs and justify defaults.

**Cache Update Threshold ($\gamma$).** Table 7 illustrates the sensitivity of our proposed method to the parameter $\gamma$. As $\gamma$ is used to control the frequency of cache updates, a consistent decrease in $\gamma$ leads to an increase in throughput. However, there is also a trend of decreasing accuracy as throughput increases. The trend is more consistent for the LLaDA-1.5 model, while for LLaDA, the accuracy at peak ($\gamma = 0.9$) is higher, but the throughput is lower.

**Sliding Window Size ($\beta$).** Fig. 3a shows that our accuracy is stable across various $\beta$ and close to No-Cache until $\beta \approx 64$; beyond that LLaDA's tendency to emit EOS early degrades results (You et al., 2025). Throughput, however, is sensitive to $\beta$: larger windows enable more parallel prediction (fewer iterations, lower latency), but overly large $\beta$ reduces cacheable MASK tokens, raising per-step compute and latency.

**Sliding Window vs. Block-Wise.** When switching Elastic-Cache to block-wise decoding (Fast-dLLM-style) (Fig. 3a), our accuracy is often similar to No-Cache, but short blocks hurt accuracy and throughput diverges. Our sliding window groups nearby MASK tokens that strongly attend to each other, whereas block-wise caching over-aggressively freezes distant MASKs, harming small-block predictions. Our Elastic-Cache's automatic cache refresh detects divergent tokens and updates them, preserving accuracy at the cost of some throughput.

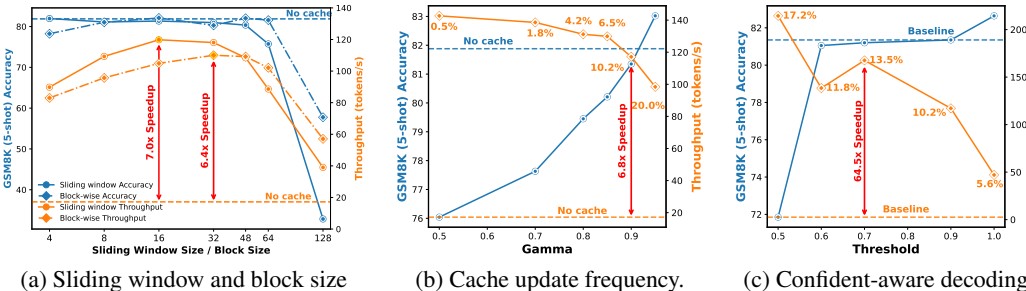

(a) Sliding window and block size     (b) Cache update frequency.     (c) Confident-aware decoding.

Figure 3: Ablation study and analysis of our proposed method. (a) Ablation study of our sliding window mechanism compared to block-wise decoding. (b) Analysis of cache update frequency under varying $\gamma$. The blue and orange lines represent accuracy and throughput, respectively. The numbers along the lines indicate the frequency of cache updates, assuming no baseline. (c) Analysis of cache update frequency under confident-aware decoding with varying $\epsilon$.

Table 6: Performance under different denoising schedules (LLaDA-1.5, GSM8K).

| Method | 1 tok/step | 2 tok/step | 4 tok/step | 3.25 tok/step ($\epsilon$=0.9) | 5.12 tok/step ($\epsilon$=0.7) |
|---|---|---|---|---|---|
| Baseline | 81.4 
 2.6 (1.0×) | 79.8 
 5.1 (1.0×) | 67.5 
 10.3 (1.0×) | 81.9 
 17.2 (1.0×) | 79.6 
 26.6 (1.0×) |
| Fast-dLLM | 80.5 
 8.5 (3.3×) | 77.3 
 15.2 (3.0×) | 64.7 
 27.3 (2.7×) | 80.8 
 36.8 (2.1×) | 80.0 
 46.9 (1.8×) |
| Elastic-Cache | **82.6** 
 47.0 (18.1×) | **78.1** 
 86.5 (17.0×) | **69.9** 
 149.8 (14.5×) | **81.4** 
 117.2 (6.8×) | **81.2** 
 167.6 (6.3×) |

**Prefill and Generation Length.** Table 8a and Table 8b provide insights into the impact of prefill length and generation length on the overall speedup. Notably, both Fast-dLLM and Elastic-Cache experience a decrease in throughput as the prefill length increases from 3-shot to 8-shot. However, Elastic-Cache exhibits a remarkable speedup and consistently high accuracy across different prefill lengths. Moreover, the throughput of Elastic-Cache increases with generation length, highlighting its unique scaling properties.

## 4.4 ANALYSIS

**Cache update frequency.** Fig. 3b and Fig. 3c illustrate the frequency of cache updates performed by Elastic-Cache under varying hyper-parameters $\gamma$ and $\epsilon$. The proposed method maintains a very low cache update frequency across different values of $\gamma$ (Fig. 3b). In extreme cases, with $\gamma = 0.95$, the cache update frequency increases to only 20% compared to the baseline without a cache. Moreover, increasing the model's confidence and accuracy (with $\epsilon$, Fig. 3c) enhances Elastic-Cache's effectiveness, and reduces the cache update frequency.

**Tunable Speed–Accuracy Trade-off.** The cache update threshold $\gamma$ directly determines the balance (Table 7). An excessively high $\gamma$ could lead to unnecessary cache updates, resulting in a decrease in speedup without any improvement in accuracy. Conversely, a smaller $\gamma$ value could guarantee speedup while sacrificing accuracy. The optimal value for $\gamma$ to maximize both accuracy and throughput depends on the model's prediction. Models with higher accuracy tend to have the best $\gamma$ value, which is closer to $1.0$ (Table 7).

**Scaling Properties.** Elastic-Cache scales greatly with the generation length and the power of the base model. Increasing the generation length slows down the baseline performance but speeds up Elastic-Cache (Tables 8b). Moreover, Elastic-Cache effectiveness is highly dependent on the accuracy of the model's predictions (Table 1, Table 3, Fig. 3c). This indicates that Elastic-Cache can effectively scale with the size of the model and the size of the training data, as LLMs generally improve when they scale up.

**Robustness Across Denoising Schedules.** We test Elastic-Cache on LLaDA-1.5, GSM8K under varying denoising schedules by controlling average tokens decoded per step (Table 6). While the baseline decodes 1 token/step, confidence-aware decoding (Wu et al., 2025) increases this to 3.25 ($\epsilon = 0.9$) and 5.12 ($\epsilon = 0.7$). As decoding becomes more aggressive, KV drift grows, demanding more frequent cache updates. Elastic-Cache adapts by raising update frequency from 5.6% to

Table 7: Impact of attention threshold on accuracy and speedup under GSM8K (5-Shot) for LLaDA and LLaDA1.5 with generation length of 512.

| Model | No Cache | Fast-dLLM | Elastic-Cache (Ours) | | | | | |
|---|---|---|---|---|---|---|---|---|
| | | | $\gamma = 0.5$ | $\gamma = 0.7$ | $\gamma = 0.8$ | $\gamma = 0.85$ | $\gamma = 0.9$ | $\gamma = 0.95$ |
| LLaDA | 77.10 | 74.83 | 71.57 | 73.46 | 74.30 | 74.68 | 77.71 | 76.72 |
| | 3.6 (1.0×) | 44.0 (12.2×) | 109.9 (30.5×) | 108.7 (30.2×) | 103.9 (28.9×) | 99.1 (27.5×) | 91.5 (25.4×) | 75.5 (21.0×) |
| LLaDA-1.5 | 81.35 | 80.82 | 76.04 | 77.63 | 79.45 | 80.21 | 81.35 | 83.02 |
| | 2.6 (1.0×) | 36.8 (14.2×) | 142.7 (54.9×) | 138.6 (53.3×) | 131.2 (50.5×) | 129.9 (50.0×) | 117.2 (45.1×) | 98.4 (37.8×) |

Table 8: Comparison between Elastic-Cache and Fast-dLLM when varying Prefill and Gen. Length.

(a) Impact of few-shots on Accuracy and Speedup Under GSM8K (1024) for LLaDA.

| Model | 3-shot | 5-shot | 8-shot |
|---|---|---|---|
| Fast-dLLM | 73.77 | 76.04 | 75.36 |
| | 28.5 (1.0×) | 25.0 (1.0×) | 20.8 (1.0×) |
| Elastic-Cache | 75.13 | 75.21 | 75.28 |
| | 185.3 (6.5×) | 169.8 (6.8×) | 143.9 (6.9×) |

(b) Impact of generation length on Accuracy and Speedup Under GSM8K (5-Shot) for LLaDA.

| Model | 256 | 512 | 1024 |
|---|---|---|---|
| Fast-dLLM | 77.94 | 74.83 | 76.04 |
| | 53.7 (1.0×) | 44.0 (1.0×) | 25.0 (1.0×) |
| Elastic-Cache | 78.24 | 77.71 | 75.21 |
| | 58.0 (1.1×) | 91.5 (2.1×) | 169.8 (6.8×) |

17.2%, preserving accuracy, unlike Fast-dLLM, which suffers under fixed schedules. This highlights Elastic-Cache's robustness to denoising variations without manual tuning.

## 5 RELATED WORK

**Diffusion Language Models.** Classical diffusion models excel in continuous domains: images (Ho et al., 2020; Dhariwal & Nichol, 2021; Rombach et al., 2022), audio (Yang et al., 2023; Huang et al., 2023), and video (Xing et al., 2024; Ho et al., 2022a;b), building on the seminal formulation of Sohl-Dickstein et al. (2015). Adapting diffusion to discrete text has followed Markov/multinomial/continuous-time paths (Austin et al., 2021a; Hoogeboom et al., 2021b;a; Campbell et al., 2022; Sun et al., 2022), refined via score matching, ratio methods, and reparameterization (Meng et al., 2022; Lou & Ermon, 2023; Zheng et al., 2023), with recent work unifying these views (Sahoo et al., 2024; Shi et al., 2024; Ou et al., 2024; Zheng et al., 2024). Early NLP systems validated these ideas (He et al., 2022; Li et al., 2022; Gong et al., 2022) and explored semi-autoregression (Han et al., 2022). Masked diffusion approaching autoregressive quality (Sahoo et al., 2024) enabled scalable models (LLaDA) competitive with LLaMA (Nie et al., 2025a; 2024; Touvron et al., 2023a; Dubey et al., 2024), with further gains from AR adaptation and instruction tuning (Gong et al., 2024; Zhu et al., 2025; Ye et al., 2025). The paradigm now spans multimodal/structured domains (You et al., 2025; Yang et al., 2025; Yu et al., 2025; Wang et al., 2024a;b; Kitouni et al., 2023).

**Acceleration Techniques for Large Language Models.** KV caching underpins efficient transformer inference (Vaswani et al., 2017; Pope et al., 2023), complemented by GQA, RoPE, and modern LLM optimizations (Ainslie et al., 2023; Su et al., 2024; Touvron et al., 2023a;b; Dubey et al., 2024). Diffusion LLMs complicate caching due to bidirectional attention and evolving representations; dedicated methods include Fast-dLLM (Wu et al., 2025), dKV-Cache (Ma et al., 2025), and DeepCache (Ma et al., 2024). Orthogonal accelerations exploit parallel/non-AR generation (Gu et al., 2017; Xiao et al., 2023), block-wise diffusion (Arriola et al., 2025), fast sampling (Chen et al., 2023), test-time scaling (Ramesh & Mardani, 2025), and consistency models (Kou et al., 2024). However, most rely on temporal heuristics or fixed thresholds, leaving attention patterns underused.

## 6 CONCLUSION

We presented **Elastic-Cache**, a training-free, architecture-agnostic policy that makes KV caching in diffusion LLMs adaptive along two axes: *when* to refresh (via an attention-aware drift test) and *where* to refresh (via a depth-selective update starting at a learned boundary layer). By block-caching distant MASK tokens, reusing shallow-layer caches, and refreshing only when the most-attended token indicates meaningful state change, Elastic-Cache removes large amounts of redundant QKV work. Across decoding steps, this yields substantial latency reductions with negligible impact on generation quality, addressing a key deployment bottleneck for diffusion decoders. Looking ahead, we plan to refine drift thresholds with learned predictors, formalize guarantees linking attention patterns to KV drift, and explore interplay with speculative decoding or other hardware-aware scheduling, extending the same principles to autoregressive LLMs and multimodal diffusion frameworks.

ETHICS STATEMENT

This work targets inference-time efficiency for diffusion LLMs and does not introduce new data collection or model training. All evaluations use publicly available datasets and third-party check-points under their original licenses, no personally identifiable information is processed. While faster decoding can lower the cost of generation and thus broaden access, it may also amplify misuse. We neither change safety filters nor attempt to bypass alignment constraints of the underlying models. We will document evaluation prompts and tasks, follow the usage policies of model providers, and encourage human oversight for downstream deployments, especially in high-stakes applications.

REPRODUCIBILITY STATEMENT

Elastic-Cache is training-free and defined by a small set of inference hyperparameters: the attention similarity threshold $\gamma$, block size and generation length. We will release code, configs, and scripts to reproduce all results: (i) reference implementations of Attention-Aware and Layer-Aware KV updates with ablation; (ii) exact prompts/datasets, metrics, and other criteria; and (iii) environment specs (CUDA/driver, framework versions) and hardware details (GPU type, batch sizes). We report wall-clock latency and accuracy metrics for each setting, and provide logs to our tables/figures from raw traces.

ACKNOWLEDGEMENTS

This research was supported by the MBZUAI-WIS Joint Program for AI Research.

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

# APPENDIX

## CONTENTS

# A  THEORETICAL VALIDATION FOR ELASTIC-CACHE

## A.1  NOTATION AND SETUP

- $L$: number of transformer layers, indexed by $\ell \in \{1, \ldots, L\}$
- $T$: total denoising steps, indexed by $t \in \{0, \ldots, T\}$
- $N$: sequence length
- $d$: hidden dimension; $d_k$, $d_v$: key and value dimensions
- $\mathbf{H}_i^{t,\ell} \in \mathbb{R}^d$: hidden state of token $i$ at step $t$, layer $\ell$
- $\mathbf{K}_i^{t,\ell}, \mathbf{V}_i^{t,\ell} \in \mathbb{R}^{d_k}, \mathbb{R}^{d_v}$: key and value of token $i$
- $\mathbf{S}^{t,\ell} \in \mathbb{R}^{N \times N}$: attention weights at step $t$, layer $\ell$
- $\mathcal{D}^{<t}$: decoded token positions up to step $t-1$
- $\mathcal{M}^t$: masked token positions at step $t$
- $\mathcal{M}_\beta^t$: sliding window of size $\beta$ over masked positions
- $\alpha_k^{t,\ell} := \sum_{q \in \mathcal{M}_\beta^t} \mathbf{S}_{q,k}^{t,\ell}$ : total attention token $k$ receives
- $\Delta\mathbf{H}_i := \mathbf{H}_i^{t,\ell} - \mathbf{H}_i^{t-1,\ell}$ : change in hidden state
- $\bar{\Delta}^{t,\ell} := \frac{1}{N}\sum_{i=1}^N \|\Delta\mathbf{H}_i^{t,\ell}\|_2$ : average hidden state drift
- $\Delta_{\max} := \max_i \|\Delta\mathbf{H}_i\|_2$ : maximum hidden state change
- $\Gamma^{t,\ell} := \alpha_{\mathcal{T}^{t,\ell}}^{t,\ell} - \max_{k \neq \mathcal{T}^{t,\ell}} \alpha_k^{t,\ell} \geq 0$ : Attention Gap

## A.2  BACKGROUND LEMMAS AND ASSUMPTIONS

**Lemma A.1** (Lipschitz Continuity of Softmax). *Based on the Proposition 2 in Gao & Pavel (2017), the softmax function $\sigma : \mathbb{R}^n \to \Delta^{n-1}$ defined by*

$$\sigma(\mathbf{z})_i = \frac{\exp(z_i)}{\sum_{j=1}^n \exp(z_j)} \tag{9}$$

*is 1-Lipschitz continuous with respect to the $\ell_2$ norm:*

$$\|\sigma(\mathbf{z}) - \sigma(\mathbf{z}')\|_2 \leq \|\mathbf{z} - \mathbf{z}'\|_2, \quad \forall \mathbf{z}, \mathbf{z}' \in \mathbb{R}^n \tag{10}$$

**Assumption A.2** (Bounded Representations). At each layer $\ell$ and step $t$: $\left\|\mathbf{H}_i^{t,\ell}\right\|_2 \leq R_\ell$

**Assumption A.3** (Lipschitz Network Components). The projection matrices satisfy $\left\|\mathbf{W}_Q^\ell\right\|_2, \left\|\mathbf{W}_K^\ell\right\|_2, \left\|\mathbf{W}_V^\ell\right\|_2 \leq W_{\max}$. The feedforward network at layer $\ell$ is $L_{\text{FFN}}$-Lipschitz continuous.

**Assumption A.4** (Progressive Unmasking). At each step $t$, a non-empty subset $\mathcal{D}^t \subseteq \mathcal{M}^{t-1}$ is unmasked: $|\mathcal{D}^{<t}|$ increases and $\mathcal{M}^t = \mathcal{M}^{t-1} \setminus \mathcal{D}^t$.

**Assumption A.5** (Layer-Wise Representation Dynamics). There exists $\ell^* \in \{1, \ldots, L\}$ and functions $f_\ell(t) \to 0$ as $t \to T$ for $\ell \leq \ell^*$ such that:

- **Shallow layers** ($\ell \leq \ell^*$): The expected hidden state change for decoded tokens vanishes: For $\ell \leq \ell^*$:
$$\mathbb{E}[\left\|\mathbf{H}_i^{t,\ell} - \mathbf{H}_i^{t-1,\ell}\right\|_2 \mid i \in \mathcal{D}^{<t}] \leq f_\ell(t) \to 0$$

- **Deep layers** ($\ell > \ell^*$): The expected change remains bounded away from zero:
$$\liminf_{t \to T} \mathbb{E}[\left\|\mathbf{H}_i^{t,\ell} - \mathbf{H}_i^{t-1,\ell}\right\|_2 \mid i \in \mathcal{D}^{<t}] \geq c_\ell > 0$$

This reflects that early layers encode local lexical patterns that stabilize quickly, while deep layers encode semantic relationships that continue evolving (Kovaleva et al., 2019; Jawahar et al., 2019; Rogers et al., 2021). Our experiments validate this (Fig. 1b).

**Assumption A.6** (Attention Concentration). The attention gap is a non-negligible fraction of total attention mass:

$$\Gamma^{t,\ell} \geq c \cdot |\mathcal{M}_\beta^t| \tag{11}$$

for some constant $c > 0$ independent of $N, t, \ell$.

**Definition A.7** (KV Drift). The KV drift at layer $\ell$, step $t$ for token $i$ is:

$$\Delta_i^{t,\ell} := \left\| \mathbf{K}_i^{t,\ell} - \mathbf{K}_i^{t-1,\ell} \right\|_2 + \left\| \mathbf{V}_i^{t,\ell} - \mathbf{V}_i^{t-1,\ell} \right\|_2 \tag{12}$$

Average drift over decoded tokens: $\Delta^{t,\ell} := \frac{1}{|\mathcal{D}^{<t}|} \sum_{i \in \mathcal{D}^{<t}} \Delta_i^{t,\ell}$

## A.3 LAYER-WISE KV DRIFT MONOTONICITY

This theorem formalizes the observation that KV drift increases with layer depth, providing theoretical justification for our layer-aware cache refresh strategy that selectively recomputes deeper layers while reusing shallow-layer caches. Fig. 1a empirically validates this monotonicity property.

**Theorem A.8** (Layer-Wise KV Drift Monotonicity). *Under Assumptions A.2–A.5, there exists a transition layer $\ell^* \in \{1, \ldots, L\}$ such that for sufficiently large $t$ (when most tokens are decoded):*

$$\mathbb{E}_t[\Delta^{t,\ell}] \leq \mathbb{E}_t[\Delta^{t,\ell'}], \quad \forall \ell \leq \ell^* < \ell' \leq L \tag{13}$$

*Proof.* **Step 1: Relating KV Drift to Hidden State Drift.**

The key-value projections at layer $\ell$ are:

$$\mathbf{K}_i^{t,\ell} = W_K^\ell \mathbf{H}_i^{t,\ell} \tag{14}$$

$$\mathbf{V}_i^{t,\ell} = W_V^\ell \mathbf{H}_i^{t,\ell} \tag{15}$$

By the triangle inequality and Assumption A.3 ($\|W_K^\ell\|_2, \|W_V^\ell\|_2 \leq W_{\max}$):

$$\begin{aligned}
\|\mathbf{K}_i^{t,\ell} - \mathbf{K}_i^{t-1,\ell}\|_2 &= \|W_K^\ell(\mathbf{H}_i^{t,\ell} - \mathbf{H}_i^{t-1,\ell})\|_2 \\
&\leq \|W_K^\ell\|_2 \|\mathbf{H}_i^{t,\ell} - \mathbf{H}_i^{t-1,\ell}\|_2 \\
&\leq W_{\max} \|\Delta \mathbf{H}_i^{t,\ell}\|_2
\end{aligned} \tag{16}$$

Similarly for values:

$$\|\mathbf{V}_i^{t,\ell} - \mathbf{V}_i^{t-1,\ell}\|_2 \leq W_{\max} \|\Delta \mathbf{H}_i^{t,\ell}\|_2 \tag{17}$$

Therefore:

$$\Delta_i^{t,\ell} = \|\mathbf{K}_i^{t,\ell} - \mathbf{K}_i^{t-1,\ell}\|_2 + \|\mathbf{V}_i^{t,\ell} - \mathbf{V}_i^{t-1,\ell}\|_2 \leq 2W_{\max}\|\Delta \mathbf{H}_i^{t,\ell}\|_2 \tag{18}$$

**Step 2: Layer Recursion for Hidden States.**

At layer $\ell$, the transformer block computes:

$$\mathbf{H}_i^{t,\ell+1} = \mathbf{H}_i^{t,\ell} + \text{Attn}^\ell(\mathbf{Q}_i^{t,\ell}, \mathbf{K}^{t,\ell}, \mathbf{V}^{t,\ell}) + \text{FFN}^\ell(\mathbf{H}_i^{t,\ell} + \text{Attn}^\ell(\cdot)) \tag{19}$$

where the attention output is:

$$\text{Attn}^\ell(\mathbf{Q}_i^{t,\ell}, \mathbf{K}^{t,\ell}, \mathbf{V}^{t,\ell}) = \sum_{j=1}^N \mathbf{S}_{i,j}^{t,\ell} \mathbf{V}_j^{t,\ell} \tag{20}$$

The change in hidden state at layer $\ell + 1$ satisfies:

$$\begin{aligned}
\|\Delta \mathbf{H}_i^{t,\ell+1}\|_2 &= \|\mathbf{H}_i^{t,\ell+1} - \mathbf{H}_i^{t-1,\ell+1}\|_2 \\
&\leq \|\Delta \mathbf{H}_i^{t,\ell}\|_2 + \|\text{Attn}^\ell(t) - \text{Attn}^\ell(t-1)\|_2 \\
&\quad + \|\text{FFN}^\ell(\text{input}^t) - \text{FFN}^\ell(\text{input}^{t-1})\|_2
\end{aligned} \tag{21}$$

By Assumption A.3, the FFN is $L_{\text{FFN}}$-Lipschitz:

$$\|\text{FFN}^\ell(\text{input}^t) - \text{FFN}^\ell(\text{input}^{t-1})\|_2 \le L_{\text{FFN}}\|\text{input}^t - \text{input}^{t-1}\|_2 \tag{22}$$

The FFN input is $\mathbf{H}_i^{t,\ell} + \text{Attn}^\ell(\cdot)$, so:

$$\|\text{input}^t - \text{input}^{t-1}\|_2 \le \|\Delta\mathbf{H}_i^{t,\ell}\|_2 + \|\text{Attn}^\ell(t) - \text{Attn}^\ell(t-1)\|_2 \tag{23}$$

Therefore:

$$\|\Delta\mathbf{H}_i^{t,\ell+1}\|_2 \le (1 + L_{\text{FFN}})\|\Delta\mathbf{H}_i^{t,\ell}\|_2 + (1 + L_{\text{FFN}})\|\text{Attn}^\ell(t) - \text{Attn}^\ell(t-1)\|_2 \tag{24}$$

**Step 3: Bounding Attention Output Change.**

Denote $\Delta_{\text{attn}}^{t,\ell,i} := \|\text{Attn}^\ell(t) - \text{Attn}^\ell(t-1)\|_2$. We decompose:

$$\sum_{j=1}^N \mathbf{S}_{i,j}^{t,\ell}\mathbf{V}_j^{t,\ell} - \sum_{j=1}^N \mathbf{S}_{i,j}^{t-1,\ell}\mathbf{V}_j^{t-1,\ell}$$
$$= \sum_{j=1}^N \mathbf{S}_{i,j}^{t,\ell}(\mathbf{V}_j^{t,\ell} - \mathbf{V}_j^{t-1,\ell}) + \sum_{j=1}^N (\mathbf{S}_{i,j}^{t,\ell} - \mathbf{S}_{i,j}^{t-1,\ell})\mathbf{V}_j^{t-1,\ell} \tag{25}$$

Taking norms and applying triangle inequality:

$$\Delta_{\text{attn}}^{t,\ell,i} \le \sum_{j=1}^N \mathbf{S}_{i,j}^{t,\ell}\|\mathbf{V}_j^{t,\ell} - \mathbf{V}_j^{t-1,\ell}\|_2 + \sum_{j=1}^N |\mathbf{S}_{i,j}^{t,\ell} - \mathbf{S}_{i,j}^{t-1,\ell}|\|\mathbf{V}_j^{t-1,\ell}\|_2 \tag{26}$$

*Step 3a: First term (value changes).* Since $\sum_j \mathbf{S}_{i,j}^{t,\ell} = 1$ (attention weights sum to 1):

$$\sum_{j=1}^N \mathbf{S}_{i,j}^{t,\ell}\|\mathbf{V}_j^{t,\ell} - \mathbf{V}_j^{t-1,\ell}\|_2 \le \sum_{j=1}^N \mathbf{S}_{i,j}^{t,\ell}W_{\max}\|\Delta\mathbf{H}_j^{t,\ell}\|_2 \quad \text{(by Assumption A.3)}$$
$$= W_{\max}\mathbb{E}_{j\sim\mathbf{S}_{i,:}^{t,\ell}}[\|\Delta\mathbf{H}_j^{t,\ell}\|_2]$$
$$\le W_{\max}\bar{\Delta}^{t,\ell} \tag{27}$$

*Step 3b: Second term (attention weight changes).* By Cauchy-Schwarz: $\sum_j |a_j|b_j \le (\sum_j |a_j|)\max_j b_j$

By Assumption A.2: $\|\mathbf{V}_j^{t-1,\ell}\|_2 \le W_{\max}R_\ell$

Therefore:

$$\sum_{j=1}^N |\mathbf{S}_{i,j}^{t,\ell} - \mathbf{S}_{i,j}^{t-1,\ell}|\|\mathbf{V}_j^{t-1,\ell}\|_2 \le W_{\max}R_\ell \sum_{j=1}^N |\mathbf{S}_{i,j}^{t,\ell} - \mathbf{S}_{i,j}^{t-1,\ell}| \tag{28}$$

By the inequality $\|\mathbf{v}\|_1 \le \sqrt{n}\|\mathbf{v}\|_2$:

$$\sum_{j=1}^N |\mathbf{S}_{i,j}^{t,\ell} - \mathbf{S}_{i,j}^{t-1,\ell}| \le \sqrt{N}\|\mathbf{S}_{i,:}^{t,\ell} - \mathbf{S}_{i,:}^{t-1,\ell}\|_2 \tag{29}$$

By Lemma A.1 (softmax is 1-Lipschitz in $\ell_2$):

$$\|\mathbf{S}_{i,:}^{t,\ell} - \mathbf{S}_{i,:}^{t-1,\ell}\|_2 \le \|\mathbf{z}_i^{t,\ell} - \mathbf{z}_i^{t-1,\ell}\|_2 \tag{30}$$

where $\mathbf{z}_i^{t,\ell} = (z_{i,1}^{t,\ell}, \ldots, z_{i,N}^{t,\ell})$ with $z_{i,j}^{t,\ell} = \frac{1}{\sqrt{d_k}}\mathbf{Q}_i^{t,\ell} \cdot \mathbf{K}_j^{t,\ell}$.

*Step 3c: Bounding logit changes.* For each component:

$$z_{i,j}^{t,\ell} - z_{i,j}^{t-1,\ell} = \frac{1}{\sqrt{d_k}}[\mathbf{Q}_i^{t,\ell} \cdot \mathbf{K}_j^{t,\ell} - \mathbf{Q}_i^{t-1,\ell} \cdot \mathbf{K}_j^{t-1,\ell}]$$

$$= \frac{1}{\sqrt{d_k}}[\mathbf{Q}_i^{t,\ell} \cdot (\mathbf{K}_j^{t,\ell} - \mathbf{K}_j^{t-1,\ell}) + (\mathbf{Q}_i^{t,\ell} - \mathbf{Q}_i^{t-1,\ell}) \cdot \mathbf{K}_j^{t-1,\ell}] \tag{31}$$

By Cauchy-Schwarz and the bounds from Assumptions A.2–A.3:

$$|z_{i,j}^{t,\ell} - z_{i,j}^{t-1,\ell}| \leq \frac{1}{\sqrt{d_k}}[W_{\max}R_\ell \cdot W_{\max}\|\Delta\mathbf{H}_j^{t,\ell}\|_2 + W_{\max}\|\Delta\mathbf{H}_i^{t,\ell}\|_2 \cdot W_{\max}R_\ell]$$

$$= \frac{W_{\max}^2 R_\ell}{\sqrt{d_k}}[\|\Delta\mathbf{H}_i^{t,\ell}\|_2 + \|\Delta\mathbf{H}_j^{t,\ell}\|_2]$$

$$\leq \frac{2W_{\max}^2 R_\ell}{\sqrt{d_k}} \max_k \|\Delta\mathbf{H}_k^{t,\ell}\|_2 \tag{32}$$

Taking $\ell_2$ norm of the logit vector:

$$\|\mathbf{z}_i^{t,\ell} - \mathbf{z}_i^{t-1,\ell}\|_2^2 = \sum_{j=1}^N |z_{i,j}^{t,\ell} - z_{i,j}^{t-1,\ell}|^2$$

$$\leq N\left(\frac{2W_{\max}^2 R_\ell}{\sqrt{d_k}}\right)^2 (\max_k \|\Delta\mathbf{H}_k^{t,\ell}\|_2)^2 \tag{33}$$

Therefore:

$$\|\mathbf{z}_i^{t,\ell} - \mathbf{z}_i^{t-1,\ell}\|_2 \leq \frac{2W_{\max}^2 R_\ell \sqrt{N}}{\sqrt{d_k}} \max_k \|\Delta\mathbf{H}_k^{t,\ell}\|_2 \tag{34}$$

For typical sequences where $\max_k \|\Delta\mathbf{H}_k^{t,\ell}\|_2 = O(\bar{\Delta}^{t,\ell})$:

$$\|\mathbf{z}_i^{t,\ell} - \mathbf{z}_i^{t-1,\ell}\|_2 \leq \frac{2W_{\max}^2 R_\ell \sqrt{N}}{\sqrt{d_k}} \bar{\Delta}^{t,\ell} \tag{35}$$

*Step 3d: Combining.* Combining the bounds from Steps 3a-3c:

$$\Delta_{\text{attn}}^{t,\ell,i} \leq W_{\max}\bar{\Delta}^{t,\ell} + W_{\max}R_\ell\sqrt{N} \cdot \frac{2W_{\max}^2 R_\ell \sqrt{N}}{\sqrt{d_k}} \bar{\Delta}^{t,\ell}$$

$$= W_{\max}\bar{\Delta}^{t,\ell}\left(1 + \frac{2W_{\max}^2 R_\ell^2 N}{\sqrt{d_k}}\right) \tag{36}$$

Define:

$$C_{\text{attn}}(\ell) := \frac{2W_{\max}^2 R_\ell^2 N}{\sqrt{d_k}} = O\left(\frac{W_{\max}^2 R_\ell^2 N}{\sqrt{d_k}}\right) \tag{37}$$

Then:

$$\Delta_{\text{attn}}^{t,\ell,i} \leq W_{\max}(1 + C_{\text{attn}}(\ell))\bar{\Delta}^{t,\ell} \tag{38}$$

**Step 4: Recursive Bound on Hidden State Drift.**

Substituting equation 38 into equation 24:

$$\|\Delta\mathbf{H}_i^{t,\ell+1}\|_2 \leq (1 + L_{\text{FFN}})\|\Delta\mathbf{H}_i^{t,\ell}\|_2 + (1 + L_{\text{FFN}})W_{\max}(1 + C_{\text{attn}}(\ell))\bar{\Delta}^{t,\ell} \tag{39}$$

Taking averages over all tokens:

$$\bar{\Delta}^{t,\ell+1} \leq [(1 + L_{\text{FFN}}) + (1 + L_{\text{FFN}})W_{\max}(1 + C_{\text{attn}}(\ell))]\bar{\Delta}^{t,\ell} \tag{40}$$

Define the layer-dependent amplification factor:

$$\lambda_\ell := (1 + L_{\text{FFN}})[1 + W_{\max}(1 + C_{\text{attn}}(\ell))] \tag{41}$$

Then:

$$\bar{\Delta}^{t,\ell+1} \leq \lambda_\ell \bar{\Delta}^{t,\ell} \tag{42}$$

**Step 5: Layer-wise Accumulation by Induction.**

By induction on $\ell$:

$$\bar{\Delta}^{t,\ell} \leq \bar{\Delta}^{t,1} \prod_{k=1}^{\ell-1} \lambda_k \tag{43}$$

Since $\lambda_\ell > 1$, drift accumulates multiplicatively across layers.

**Step 6: Applying Layer-Wise Specialization.**

By Assumption A.5:

- **Shallow layers** ($\ell \leq \ell^*$): $\bar{\Delta}^{t,\ell} \leq f_\ell(t) \to 0$ as $t \to T$

- **Deep layers** ($\ell > \ell^*$): $\liminf_{t \to T} \bar{\Delta}^{t,\ell} \geq c_\ell > 0$

By equation 18:

$$\mathbb{E}[\Delta^{t,\ell}] = \mathbb{E}\left[\frac{1}{|\mathcal{D}^{<t}|} \sum_{i \in \mathcal{D}^{<t}} \Delta_i^{t,\ell}\right] \leq 2W_{\max}\bar{\Delta}^{t,\ell} \tag{44}$$

Therefore, for sufficiently large $t$ and any $\ell \leq \ell^* < \ell'$:

$$\mathbb{E}[\Delta^{t,\ell}] \leq 2W_{\max}f_\ell(t) \to 0 \tag{45}$$

$$\mathbb{E}[\Delta^{t,\ell'}] \geq 2W_{\max}c_{\ell'} > 0 \tag{46}$$

This establishes:

$$\mathbb{E}[\Delta^{t,\ell}] < \mathbb{E}[\Delta^{t,\ell'}], \quad \forall \ell \leq \ell^* < \ell' \tag{47}$$

$$\square$$

## A.4 ATTENTION CONCENTRATION AND DRIFT

**Theorem A.9** (Attention Concentration and Drift)**.** *Let $\mathcal{T}^{t,\ell} = \arg\max_{k \in \mathcal{D}^{<t}} \sum_{q \in \mathcal{M}_\beta^t} \mathbf{S}_{q,k}^{t,\ell}$ be the most-attended token at layer $\ell$, step $t$. Under Assumptions A.2–A.3, the most-attended token has drift bounded by:*

$$\Delta_{\mathcal{T}^{t,\ell}}^{t,\ell} \leq \bar{\Delta}^{t,\ell} + \epsilon_t \tag{48}$$

*where $\bar{\Delta}^{t,\ell} = \frac{1}{|\mathcal{D}^{<t}|} \sum_{i \in \mathcal{D}^{<t}} \Delta_i^{t,\ell}$ is the average drift and $\epsilon_t = O\left(\frac{\sqrt{d_k}}{R_\ell \sqrt{N}}\right)$.*

*Proof.* **Step 1: Bounding Attention Weight Changes.**

We derive how attention weights $\mathbf{S}_{q,k}^{t,\ell}$ change when hidden states change.

*Step 1a: Logit change.* The attention logits are $z_{q,k} = \frac{1}{\sqrt{d_k}}\mathbf{Q}_q \cdot \mathbf{K}_k$ where:

$$\mathbf{Q}_q = W_Q \mathbf{H}_q, \quad \mathbf{K}_k = W_K \mathbf{H}_k \tag{49}$$

The change in logits between steps $t$ and $t-1$ is:

$$z_{q,k}^{t,\ell} - z_{q,k}^{t-1,\ell} = \frac{1}{\sqrt{d_k}}[\mathbf{Q}_q^{t,\ell} \cdot \mathbf{K}_k^{t,\ell} - \mathbf{Q}_q^{t-1,\ell} \cdot \mathbf{K}_k^{t-1,\ell}] \tag{50}$$

Using the identity $ab - a'b' = a(b - b') + (a - a')b'$:

$$= \frac{1}{\sqrt{d_k}}[\mathbf{Q}_q^{t,\ell} \cdot (\mathbf{K}_k^{t,\ell} - \mathbf{K}_k^{t-1,\ell}) + (\mathbf{Q}_q^{t,\ell} - \mathbf{Q}_q^{t-1,\ell}) \cdot \mathbf{K}_k^{t-1,\ell}] \tag{51}$$

*Step 1b: Apply Cauchy-Schwarz inequality.* Taking absolute value and applying Cauchy-Schwarz:

$$|z_{q,k}^{t,\ell} - z_{q,k}^{t-1,\ell}| \leq \frac{1}{\sqrt{d_k}}[\|\mathbf{Q}_q^{t,\ell}\|_2 \|\mathbf{K}_k^{t,\ell} - \mathbf{K}_k^{t-1,\ell}\|_2$$
$$+ \|\mathbf{Q}_q^{t,\ell} - \mathbf{Q}_q^{t-1,\ell}\|_2 \|\mathbf{K}_k^{t-1,\ell}\|_2] \tag{52}$$

*Step 1c: Bound projection norms.* By Assumption A.2: $\|\mathbf{H}_i^{t,\ell}\|_2 \leq R_\ell$ for all $i, t$.

By Assumption A.3: $\|W_Q\|_2, \|W_K\|_2 \leq W_{\max}$.

Therefore:

$$\|\mathbf{Q}_q^{t,\ell}\|_2 \leq \|W_Q\|_2 \|\mathbf{H}_q^{t,\ell}\|_2 \leq W_{\max} R_\ell \tag{53}$$

$$\|\mathbf{K}_k^{t,\ell}\|_2 \leq \|W_K\|_2 \|\mathbf{H}_k^{t,\ell}\|_2 \leq W_{\max} R_\ell \tag{54}$$

$$\|\mathbf{K}_k^{t,\ell} - \mathbf{K}_k^{t-1,\ell}\|_2 \leq \|W_K\|_2 \|\mathbf{H}_k^{t,\ell} - \mathbf{H}_k^{t-1,\ell}\|_2 \leq W_{\max} \|\Delta\mathbf{H}_k\|_2 \tag{55}$$

$$\|\mathbf{Q}_q^{t,\ell} - \mathbf{Q}_q^{t-1,\ell}\|_2 \leq W_{\max} \|\Delta\mathbf{H}_q\|_2 \tag{56}$$

Substituting these bounds:

$$|z_{q,k}^{t,\ell} - z_{q,k}^{t-1,\ell}| \leq \frac{1}{\sqrt{d_k}}[W_{\max} R_\ell \cdot W_{\max} \|\Delta\mathbf{H}_k\|_2 + W_{\max} \|\Delta\mathbf{H}_q\|_2 \cdot W_{\max} R_\ell]$$
$$= \frac{W_{\max}^2 R_\ell}{\sqrt{d_k}}[\|\Delta\mathbf{H}_k\|_2 + \|\Delta\mathbf{H}_q\|_2] \tag{57}$$

*Step 1d: Use maximum drift.* Since $\|\Delta\mathbf{H}_i\|_2 \leq \Delta_{\max}$ for all $i$:

$$|z_{q,k}^{t,\ell} - z_{q,k}^{t-1,\ell}| \leq \frac{2W_{\max}^2 R_\ell}{\sqrt{d_k}} \Delta_{\max} \tag{58}$$

*Step 1e: Compute $\ell_2$ norm of logit vector.* The logit vector for query $q$ is $\mathbf{z}_q = (z_{q,1}, \ldots, z_{q,N}) \in \mathbb{R}^N$.

By the previous bound applied to each component:

$$\|\mathbf{z}_q^{t,\ell} - \mathbf{z}_q^{t-1,\ell}\|_2^2 = \sum_{k=1}^{N} |z_{q,k}^{t,\ell} - z_{q,k}^{t-1,\ell}|^2$$
$$\leq \sum_{k=1}^{N} \left(\frac{2W_{\max}^2 R_\ell}{\sqrt{d_k}}\right)^2 \Delta_{\max}^2$$
$$= N \cdot \frac{4W_{\max}^4 R_\ell^2}{d_k} \Delta_{\max}^2 \tag{59}$$

Taking square root:

$$\|\mathbf{z}_q^{t,\ell} - \mathbf{z}_q^{t-1,\ell}\|_2 \leq \frac{2W_{\max}^2 R_\ell \sqrt{N}}{\sqrt{d_k}} \Delta_{\max} \tag{60}$$

*Step 1f: Apply softmax Lipschitz property.* By Lemma A.1 (softmax is 1-Lipschitz in $\ell_2$ norm):

$$\|\mathbf{S}_{q,:}^{t,\ell} - \mathbf{S}_{q,:}^{t-1,\ell}\|_2 \leq \|\mathbf{z}_q^{t,\ell} - \mathbf{z}_q^{t-1,\ell}\|_2 \leq \frac{2W_{\max}^2 R_\ell \sqrt{N}}{\sqrt{d_k}} \Delta_{\max} \tag{61}$$

*Step 1g: Convert to $\ell_\infty$ norm.* Since $\|\mathbf{v}\|_\infty \leq \|\mathbf{v}\|_2$ for any vector $\mathbf{v}$:

$$\max_k |\mathbf{S}_{q,k}^{t,\ell} - \mathbf{S}_{q,k}^{t-1,\ell}| \leq \frac{2W_{\max}^2 R_\ell \sqrt{N}}{\sqrt{d_k}} \Delta_{\max} \tag{62}$$

**Step 2: Change in Total Attention Received.**

For token $k$, the change in total attention received is:

$$\begin{aligned}
|\alpha_k^{t,\ell} - \alpha_k^{t-1,\ell}| &= \left| \sum_{q \in \mathcal{M}_\beta^t} (\mathbf{S}_{q,k}^{t,\ell} - \mathbf{S}_{q,k}^{t-1,\ell}) \right| \\
&\leq \sum_{q \in \mathcal{M}_\beta^t} |\mathbf{S}_{q,k}^{t,\ell} - \mathbf{S}_{q,k}^{t-1,\ell}| \quad \text{(triangle inequality)} \\
&\leq |\mathcal{M}_\beta^t| \cdot \max_q \max_k |\mathbf{S}_{q,k}^{t,\ell} - \mathbf{S}_{q,k}^{t-1,\ell}| \quad \text{(bound by max)}
\end{aligned} \tag{63}$$

Using equation 62:

$$|\alpha_k^{t,\ell} - \alpha_k^{t-1,\ell}| \leq |\mathcal{M}_\beta^t| \cdot \frac{2W_{\max}^2 R_\ell \sqrt{N}}{\sqrt{d_k}} \Delta_{\max} \tag{64}$$

**Step 3: Relating to KV Drift.**

Recall that KV drift is $\Delta_i^{t,\ell} = \|\mathbf{K}_i^{t,\ell} - \mathbf{K}_i^{t-1,\ell}\|_2 + \|\mathbf{V}_i^{t,\ell} - \mathbf{V}_i^{t-1,\ell}\|_2$.

By Assumption A.3:

$$\Delta_i^{t,\ell} \leq W_{\max}\|\Delta \mathbf{H}_i\|_2 + W_{\max}\|\Delta \mathbf{H}_i\|_2 = 2W_{\max}\|\Delta \mathbf{H}_i\|_2 \tag{65}$$

Therefore: $\|\Delta \mathbf{H}_i\|_2 \geq \frac{\Delta_i^{t,\ell}}{2W_{\max}}$.

In particular: $\Delta_{\max} \geq \frac{\max_i \Delta_i^{t,\ell}}{2W_{\max}}$.

Substituting into equation 64:

$$|\alpha_k^{t,\ell} - \alpha_k^{t-1,\ell}| \leq |\mathcal{M}_\beta^t| \cdot \frac{2W_{\max}^2 R_\ell \sqrt{N}}{\sqrt{d_k}} \cdot \frac{\max_i \Delta_i^{t,\ell}}{2W_{\max}} = |\mathcal{M}_\beta^t| \cdot \frac{W_{\max} R_\ell \sqrt{N}}{\sqrt{d_k}} \max_i \Delta_i^{t,\ell} \tag{66}$$

**Step 4: Stability Constraint and Excess Drift.**

Suppose $\mathcal{T}^{t,\ell}$ has drift $\Delta_{\mathcal{T}^{t,\ell}}^{t,\ell} = \bar{\Delta}^{t,\ell} + \varepsilon$ where $\varepsilon > 0$ is excess drift beyond average.

Then:

$$|\alpha_{\mathcal{T}^{t,\ell}}^{t,\ell} - \alpha_{\mathcal{T}^{t,\ell}}^{t-1,\ell}| \leq |\mathcal{M}_\beta^t| \cdot \frac{W_{\max} R_\ell \sqrt{N}}{\sqrt{d_k}} (\bar{\Delta}^{t,\ell} + \varepsilon) \tag{67}$$

While tokens with average drift have:

$$|\alpha_k^{t,\ell} - \alpha_k^{t-1,\ell}| \leq |\mathcal{M}_\beta^t| \cdot \frac{W_{\max} R_\ell \sqrt{N}}{\sqrt{d_k}} \bar{\Delta}^{t,\ell} \tag{68}$$

The differential attention change is:

$$\Delta_{\text{differential}} = |\mathcal{M}_\beta^t| \cdot \frac{W_{\max} R_\ell \sqrt{N}}{\sqrt{d_k}} \varepsilon \tag{69}$$

For $\mathcal{T}^{t,\ell}$ to remain most-attended, the gap at step $t-1$ must absorb this differential:

$$\Gamma^{t-1,\ell} \geq \Delta_{\text{differential}} = |\mathcal{M}_\beta^t| \cdot \frac{W_{\max} R_\ell \sqrt{N}}{\sqrt{d_k}} \varepsilon \tag{70}$$

**Step 5: Assuming Bounded Attention Gap.**

Applying the assumption A.6:

$$c \cdot |\mathcal{M}_\beta^t| \geq |\mathcal{M}_\beta^t| \cdot \frac{W_{\max} R_\ell \sqrt{N}}{\sqrt{d_k}} \varepsilon \tag{71}$$

Canceling $|\mathcal{M}_\beta^t|$ (assuming $|\mathcal{M}_\beta^t| > 0$):

$$c \geq \frac{W_{\max} R_\ell \sqrt{N}}{\sqrt{d_k}} \varepsilon \tag{72}$$

Solving for $\varepsilon$:

$$\varepsilon \leq \frac{c\sqrt{d_k}}{W_{\max} R_\ell \sqrt{N}} = O\left(\frac{\sqrt{d_k}}{R_\ell \sqrt{N}}\right) \tag{73}$$

Therefore:

$$\Delta_{\mathcal{T}^{t,\ell}}^{t,\ell} \leq \bar{\Delta}^{t,\ell} + O\left(\frac{\sqrt{d_k}}{R_\ell \sqrt{N}}\right) \tag{74}$$

$\square$

### A.5 Implications for Elastic-Cache

These results provide theoretical justification for our design:

- **Theorem A.8**: Deeper layers have larger KV drift, justifying layer-aware refresh starting from $\ell^*$
- **Theorem A.9**: Most-attended tokens have minimal drift, validating their use as cache staleness indicators

## B Detailed Experiment Setup

**Implementation Details.** We conduct all the experiments on a single NVIDIA A100 80GB GPU to ensure a consistent hardware environment. We evaluate our proposed method, **Elastic-Cache**, on three large scale DLMs: LLaDA-Instruct (Nie et al., 2025a), LLaDA-1.5 (Zhu et al., 2025), and the multimodal LLaDA-V (You et al., 2025). Our evaluation spans both language and multimodal reasoning tasks including MBPP (Austin et al., 2021b), HumanEval (Chen et al., 2021) for coding tasks, MATH (Hendrycks et al., 2021), GSM8K (Cobbe et al., 2021) for Maths related tasks and MathVista (Lu et al., 2023) MathVerse (Zhang et al., 2024b) for multimodal mathematical reasoning tasks. The major hyperparameters for Elastic-Cache, unless otherwise specified in ablation studies, are set to a attention threshold of $\gamma = 0.9$, a confidence threshold for parallel decoding of $\epsilon = 0.9$, and a cache block size of 32. To establish a rigorous and fair comparison for all baseline methods, were re-evaluate all the methods including the original diffusion model LLaDA Nie et al. (2025a) and Fast-dLLM (Wu et al., 2025). This process eliminates confounding variables from hardware or software discrepancies and ensures that all observed performance differences are attributable to the methods themselves.

**Evaluation Framework and Metrics.** Our evaluation protocol comprehensively assesses both inference efficiency and the preservation of model performance across a variety of tasks. For standardization and reproducibility, we conduct all task-specific evaluations using the `lm-eval-harness` library (Gao et al., 2024). We measure inference speed by throughput in tokens per second (t/s), which we calculate as the average number of tokens the model generates over the entire sequence until it produces an end-of-sequence (`<eos>`) token. We keep our calculation methodology consistent with that of Fast-dLLM (Wu et al., 2025) to ensure comparable speed benchmarks. We measure task-specific performance using established metrics appropriate for each benchmark: for GSM8K (Cobbe et al., 2021), we report 5-shot `flexible_extract` exact match accuracy; for the MATH dataset (Hendrycks et al., 2021), we report the 4-shot `math_verify` score using the `minerva_math` variant; for HumanEval (Chen et al., 2021), we evaluate 0-shot accuracy using

Table 9: The hyper-parameters of Elastic-Cache under various settings.

| Model | Benchmark | Gen Length | $\beta$ | $\gamma$ |
|---|---|---|---|---|
| LLaDA | GSM8K (5-shot) | 256 | 32 | 0.9 |
| | | 512 | 16 | 0.9 |
| | MATH (4-shot) | 256 | 16 | 0.9 |
| | | 512 | 16 | 0.9 |
| | Humaneval (0-shot) | 256 | 32 | 0.9 |
| | | 512 | 32 | 0.9 |
| | MBPP (3-shot) | 256 | 16 | 0.9 |
| | | 512 | 16 | 0.9 |
| LLaDA-1.5 | GSM8K (5-shot) | 256 | 16 | 0.9 |
| | | 512 | 16 | 0.9 |
| | MATH (4-shot) | 256 | 16 | 0.9 |
| | | 512 | 16 | 0.9 |
| | Humaneval (0-shot) | 256 | 32 | 0.9 |
| | | 512 | 32 | 0.9 |
| | MBPP (3-shot) | 256 | 16 | 0.9 |
| | | 512 | 16 | 0.9 |
| LLaDA-V | Mathvista | 256 | 16 | 0.7 |
| | | 512 | 16 | 0.7 |
| | Mathverse | 256 | 16 | 0.7 |
| | | 512 | 16 | 0.7 |

a post-processing script consistent with the Fast-dLLM implementation to ensure fair comparison; and for MBPP (Austin et al., 2021b), we report the 3-shot `pass@1` metric. For multimodal evaluation on LLaDA-V (You et al., 2025), we utilize an evaluation suite adapted from its official implementation using the `lmms-eval` framework (Zhang et al., 2024a; Li et al., 2024) to test on the MathVista (Lu et al., 2023) and MathVerse (Zhang et al., 2024b) benchmarks. For MathVista, we report the `gpt_eval_score`, and for MathVerse, we report the `gpt_eval_score` on the `mathverse_testmini_vision_dominant` subset.

**Hyper-parameters:** The hyper-parameters used for Elastic-Cache are provided in Table 9. Specifically,

- For LLaDA and LLaDA-1.5, $\gamma = 0.9$ everywhere; $\beta$ is mostly 16, except GSM8K ($\beta = 32$ at 256, 16 at 512) and HumanEval ($\beta = 32$ at both 256/512).
- For LLaDA-V (MathVista/MathVerse), $\gamma = 0.7$ and $\beta = 16$ for both 256 and 512 token lengths.
- All tasks are reported at generation lengths 256 and 512.

**Our Motivation and Perspective.** We close the gap with attention-aware and layer-aware caching for diffusion LLMs: tracking most-attended tokens and depth-varying KV dynamics to guide recomputation, complementary to interval-based (Ma et al., 2025) and confidence-based (Wu et al., 2025) policies and compatible with the broader acceleration toolkit (Ainslie et al., 2023; Su et al., 2024; Touvron et al., 2023a;b; Dubey et al., 2024; Gu et al., 2017; Xiao et al., 2023; Arriola et al., 2025; Chen et al., 2023; Ramesh & Mardani, 2025; Kou et al., 2024).

## C  EXTENDED EXPERIMENTAL ANALYSIS

### C.1  COMPREHENSIVE HYPERPARAMETER SENSITIVITY

We provide extensive ablation studies to understand the interaction between window size $\beta$ and attention threshold $\gamma$. Table 10 presents results on GSM8K with 512 Gen Length with LLaDA-1.5, systematically varying both parameters. The results demonstrate that $\beta = 16$ with $\gamma = 0.9$ provides the best balance between accuracy and throughput for most applications. When maximum accuracy is required, $\beta = 8$ with $\gamma = 0.95$ achieves 83.2% accuracy at 79.1 t/s. For throughput-critical

deployments, $\beta = 16$ with $\gamma = 0.7$ delivers 138.6 t/s while maintaining 77.6% accuracy. Larger window sizes ($\beta = 32$) do not consistently improve performance, likely because they cache too many MASK tokens that eventually become relevant, forcing more frequent cache invalidations.

Table 10: Joint sensitivity analysis of window size $\beta$ and attention threshold $\gamma$ on GSM8K with 512 Gen Length (LLaDA-1.5). Each cell shows accuracy (top) and throughput in t/s (bottom).

| $\beta$ / $\gamma$ | 0.7 | 0.8 | 0.9 | 0.95 |
|---|---|---|---|---|
| 8 | 79.8 | 81.3 | 81.1 | 83.2 |
| | 103.9 | 103.5 | 109.3 | 79.1 |
| 16 | 77.6 | 79.5 | 81.4 | 83.0 |
| | 138.6 | 131.2 | 117.2 | 98.4 |
| 32 | 77.5 | 77.7 | 81.0 | 81.7 |
| | 118.7 | 116.6 | 104.1 | 88.8 |

Beyond single token tracking, we evaluate strategies that monitor the top-k most-attended tokens per layer. Table 11 shows results for k $\in \{1, 5, 10, 15, 20\}$ across different $\gamma$ values. Tracking more tokens improves accuracy slightly but adds overhead. Top-10 and Top-15 strategies achieve the best accuracy, reaching up to 84.7% on GSM8K with 512 Gen Length. However, the throughput gains diminish as more tokens require drift computation. For most deployments, tracking a single most-attended token (Top-1) provides sufficient signal while minimizing overhead.

Table 11: Sensitivity to the number of tracked tokens (Top-k) across different attention thresholds on GSM8K with 512 Gen Length (LLaDA-1.5). Each cell shows accuracy (top) and throughput in t/s (bottom).

| Top-k / $\gamma$ | 0.8 | 0.85 | 0.9 | 0.95 |
|---|---|---|---|---|
| Top-1 | 79.5 | 80.2 | 81.4 | 83.0 |
| | 131.2 | 129.9 | 117.2 | 98.4 |
| Top-5 | 81.5 | 81.4 | 83.5 | 82.7 |
| | 130.5 | 122.4 | 109.9 | 88.0 |
| Top-10 | 81.4 | 82.6 | 84.1 | 82.9 |
| | 121.1 | 118.4 | 100.7 | 77.6 |
| Top-15 | 82.5 | 83.2 | 83.7 | 84.7 |
| | 167.4 | 159.7 | 139.4 | 103.0 |
| Top-20 | 81.6 | 82.3 | 83.9 | 84.2 |
| | 162.4 | 154.1 | 136.3 | 102.0 |

## C.2 MEMORY AND COMPUTATIONAL OVERHEAD

**Memory Footprint Analysis.** Table 12 reports peak GPU memory usage across generation lengths on LLaDA-1.5 with GSM8K. Elastic-Cache achieves lower memory consumption than both baselines, with savings of 0.93-1.42 GB compared to the baseline and 2.38-3.89 GB compared to Fast-dLLM. This reduction stems from: (1) selective layer-wise cache updates that avoid storing intermediate states for all layers, and (2) block-wise caching of distant MASK tokens outside the sliding window. These memory savings enable deployment on resource-constrained devices while maintaining high throughput.

Table 12: Peak GPU memory footprint (GB) on LLaDA-1.5, GSM8K across generation lengths.

| Method | 256 tokens | 512 tokens | 1024 tokens |
|---|---|---|---|
| Baseline | 19.04 | 19.62 | 20.79 |
| Fast-dLLM | 20.49 | 21.42 | 23.26 |
| Elastic-Cache | **18.11** | **18.13** | **19.37** |

The computational overhead of our attention-aware cache update mechanism is minimal compared to standard attention computation. Table 13 compares the complexity and multiply-accumulate operations (MACs) for cache update triggers versus full QKV attention at sequence length K=1024. Finding the most-attended token requires $O(K^2 H)$ operations, which translates to $6.7 \times 10^7$ MACs. In contrast, full attention computation scales as $O(K^2 HD)$ with $1.3 \times 10^{10}$ MACs. Our cache

trigger introduces less than 0.5% overhead relative to attention, making it negligible in the overall inference budget. The cosine similarity computation for drift detection adds another $O(KH)$ operations, which is even cheaper.

Table 13: Computational overhead comparison. Cache update trigger has negligible cost compared to full attention computation (K=1024, H=32, D=128).

| Operation | Complexity | MACs (K=1024) |
|---|---|---|
| Cache Update Trigger | $O(K^2H)$ | $6.7 \times 10^7$ |
| Attention (QKV) | $O(K^2HD)$ | $1.3 \times 10^{10}$ |

## C.3 SCALABILITY ANALYSIS

We validate multi-GPU scalability by comparing throughput and latency under different hardware configurations. Table 14 shows results for LLaDA-1.5 on GSM8K with 512 Gen Length using 1 and 2 A100 GPUs with data parallelism. With 2 GPUs and batch size 8, Elastic-Cache achieves 225.5 t/s compared to Fast-dLLM's 68.0 t/s, maintaining the 3.3x throughput advantage observed in single-GPU settings. The latency improvement is more dramatic: our method reduces end-to-end inference time from 1.86 hours to 0.83 hours on a single GPU, and from 1.00 hours to 0.56 hours on two GPUs. These results confirm that Elastic-Cache scales effectively with additional hardware without requiring architecture-specific optimizations.

Table 14: Multi-GPU scalability on GSM8K with 512 Gen Length (LLaDA-1.5). Throughput in t/s and latency in hours for full benchmark evaluation.

| Configuration | Accuracy (%) | Throughput (t/s) | Latency (h) |
|---|---|---|---|
| *1 GPU, batch size 4* | | | |
| Fast-dLLM | 80.3 | 36.8 | 1.86 |
| Elastic-Cache | **81.9** | **117.2** | **0.83** |
| *2 GPUs, batch size 8* | | | |
| Fast-dLLM | 80.3 | 68.0 | 1.00 |
| Elastic-Cache | **81.9** | **225.5** | **0.56** |

## C.4 BLOCK-CACHING MECHANISM

To validate the effectiveness of our block-wise caching strategy for distant MASK tokens, we compare Elastic-Cache with and without this mechanism across different window sizes. Table 15 shows that block-caching provides substantial throughput gains with minimal impact on accuracy. At $\beta = 16$, removing block-caching reduces throughput from 119.8 t/s to 82.7 t/s while maintaining similar accuracy (81.4% vs 80.6%). The benefits increase at larger window sizes, demonstrating that caching distant MASK tokens effectively eliminates redundant computation without harming prediction quality.

Table 15: Ablation of block-caching mechanism on GSM8K with 512 Gen Length (LLaDA-1.5, $\gamma = 0.9$). Each cell shows accuracy (top) and throughput in t/s (bottom).

| Method | $\beta = 8$ | $\beta = 16$ | $\beta = 32$ | $\beta = 64$ |
|---|---|---|---|---|
| w/o block-caching | 81.1 | 80.6 | 80.6 | 74.3 |
| | 77.0 | 82.7 | 84.3 | 67.1 |
| Elastic-Cache | 81.1 | 81.4 | 81.0 | 75.7 |
| | 109.3 | 119.8 | 118.1 | 88.6 |

We further explore integrating adaptive block sizing using AdaBlock (Lu et al., 2025), which dynamically adjusts window size based on semantic coherence. Table 16 shows results when combining AdaBlock with Elastic-Cache. Starting from default window sizes $\beta_0$, AdaBlock adjusts to average sizes $\bar{\beta}$ during decoding. However, this adaptive approach introduces overhead without accuracy gains, confirming our observation that fixed window sizes suffice when combined with attention-aware cache updates.

Table 16: Integration with AdaBlock adaptive window sizing on GSM8K with 512 Gen Length (LLaDA-1.5, $\gamma = 0.9$). Format: accuracy / throughput (t/s).

| Method | $\beta_0 = 16, \bar{\beta} = 15.6$ | $\beta_0 = 32, \bar{\beta} = 27.1$ |
|---|---|---|
| Elastic-Cache | 81.4 / 119.1 | 81.0 / 118.1 |
| + AdaBlock | 81.9 / 87.1 | 80.7 / 85.5 |

## C.5 RUNTIME ADAPTATION OF LAYER BOUNDARY

The layer boundary $\ell^*$ is determined automatically at runtime based on observed attention drift. Table 17 shows how cache update frequency $\rho = \frac{L - \ell^* - 1}{L}$ varies with threshold $\gamma$ on GSM8K with 512 Gen Length. Lower $\gamma$ values trigger updates less frequently ($\rho = 0.47\%$ at $\gamma = 0.5$), maximizing throughput but sacrificing accuracy. Higher $\gamma$ values increase update frequency ($\rho = 20.02\%$ at $\gamma = 0.95$), preserving accuracy at reduced throughput. This adaptive behavior demonstrates that $\ell^*$ effectively responds to input difficulty without manual tuning.

Table 17: Cache update frequency $\rho$ and performance as layer boundary $\ell^*$ adapts to different thresholds on GSM8K with 512 Gen Length (LLaDA-1.5).

| $\gamma$ | 0.5 | 0.7 | 0.8 | 0.85 | 0.9 | 0.95 |
|---|---|---|---|---|---|---|
| $\rho$ (%) | 0.47 | 1.81 | 4.17 | 6.50 | 10.23 | 20.02 |
| Accuracy (%) | 76.0 | 77.6 | 79.5 | 80.2 | 81.4 | 83.0 |
| Throughput (t/s) | 142.7 | 138.6 | 131.2 | 129.9 | 117.2 | 98.4 |

## C.6 VALIDATION OF MOST-ATTENDED TOKEN HEURISTIC

Our method relies on the assumption that most-attended tokens exhibit minimal drift and serve as conservative indicators for cache staleness. Table 18 validates this assumption empirically by measuring average cosine similarity between consecutive steps for most-attended tokens versus all cached tokens across benchmarks. The most-attended tokens consistently maintain higher similarity (0.948-0.985), confirming they change less than average tokens and provide reliable lower bounds for drift detection.

Table 18: Empirical validation of most-attended token stability. Higher cosine similarity indicates lower drift. Results on LLaDA-1.5 with $\gamma = 0.9$.

| Token Type | GSM8K | MATH | HumanEval | MBPP |
|---|---|---|---|---|
| Most-attended | 0.974 | 0.978 | 0.985 | 0.948 |
| Average (all cached) | 0.973 | 0.977 | 0.980 | 0.947 |

## C.7 MULTIMODAL EXTENSIONS

For multimodal tasks on LLaDA-V, we evaluate two configurations: single-token prediction per step (matching our text-only setup) and parallel multi-token prediction (matching the original LLaDA-V implementation). Table 19 shows that Elastic-Cache with single-token prediction already surpasses the original LLaDA-V baseline in throughput. When extended to parallel prediction, throughput increases further to 44.2 t/s on MathVista and 42.2 t/s on MathVerse, demonstrating that our cache management strategy complements parallel decoding effectively in multimodal settings.

## C.8 COMPARISON WITH CONSISTENCY MODELS

We compare against consistency-based acceleration methods for diffusion models. Table 20 shows results on GSM8K with 512 Gen Length using Consistency LLMs (Kou et al., 2024) versus our approach on LLaDA-1.5. While consistency models achieve reasonable throughput (35.5 t/s), they suffer significant accuracy degradation (56.5% vs 78.5% baseline). Elastic-Cache maintains higher accuracy (79.2%) while delivering substantially greater throughput (109.6 t/s), demonstrating the advantage of training-free cache management over distillation-based approaches.

Table 19: Multimodal performance with parallel decoding on LLaDA-V (generation length 512). Format: accuracy / throughput (t/s).

| Method | MathVista | MathVerse |
|---|---|---|
| LLaDA-V baseline | 53.0 / 1.9 | 26.9 / 2.0 |
| Elastic-Cache (single-token) | 55.8 / 24.1 | 29.2 / 32.3 |
| Elastic-Cache (parallel) | 50.6 / 44.2 | 26.3 / 42.2 |

Table 20: Comparison with consistency-based acceleration on GSM8K with 512 *Gen Length*. Our training-free approach preserves accuracy while achieving higher throughput.

| Method | Accuracy (%) | Throughput (t/s) |
|---|---|---|
| Consistency LLMs | 56.5 | 35.5 |
| LLaDA-1.5 baseline | 78.5 | 2.6 |
| Elastic-Cache | **79.2** | **109.6** |

## D  USE OF LARGE LANGUAGE MODELS

An LLM was used to help solely polish the writing of the paper, while all method, idea and experiments were prepared and carried out entirely by the authors.

## E  SAMPLE RESPONSE

In the following, we present several examples of actual generation results produced by our approach under different parameter settings.

Table 21: Example – Qualitative comparison for different $\gamma$, LLaDA-1.5

---

**Prompt:** *A store sells 3 types of fruit: apples at $2 each, bananas at $1 each, and oranges at $3 each. If Sarah buys 4 apples, 6 bananas, and 2 oranges, how much does she spend in total?*

---

| $\gamma = 0.5$ | $\gamma = 0.7$ | $\gamma = 0.95$ |
|---|---|---|
| To determine how much Sarah spends in total, we need to calculate the cost of each type of fruit separately and then sum these amounts. 1. **Cost of apples:** - Sarah buys 4 apples. - Each apple costs $2. - Total cost for apples = $(4 \times 2 = 8)$ dollars. 2. **Cost of bananas:** - Sarah buys 6 bananas. - Each banana costs $1. - Total cost for bananas = $(6 \times 1 = 6)$ dollars. 3. **Cost of oranges:** - Sarah buys 2 oranges. - Each orange costs $3. - Total cost for oranges = $(2 \times 3 = 6)$ dollars. Now, we sum the costs of all the fruits to find the total amount Sarah spends: 8 (apples) + 6 (bananas) + 6 (oranges) = 20 dollars Therefore, Sarah spends a total of 20 dollars. **Time: 3.1s** | To determine how much Sarah spends in total, we need to calculate the cost of each type of fruit separately and then sum these amounts. First, let's calculate the cost of the apples: - Sarah buys 4 apples. - Each apple costs $2. - Therefore, the cost of apples is $(4 \times 2 = 8)$ dollars. Next, let's calculate the cost of the bananas: - Sarah buys 6 bananas. - Each banana costs $1. - Therefore, the cost of bananas is $(6 \times 1 = 6)$ dollars. Finally, let's calculate the cost of the oranges: - Sarah buys 2 oranges. - Each orange costs $3. - Therefore, the cost of oranges is $(2 \times 3 = 6)$ dollars. Now, we sum the costs of all the fruits to find the total amount Sarah spends: - Total cost = Cost of apples + Cost of bananas + Cost of oranges - Total cost = $(8 + 6 + 6 = 20)$ dollars So, Sarah spends a total of 20 dollars. **Time: 3.4s** | To determine how much Sarah spends in total, we need to calculate the cost of each type of fruit separately and then sum these amounts. First, let's calculate the cost of the apples: - Sarah buys 4 apples. - Each apple costs $2. - Total cost for apples = $(4 \times 2 = 8)$ dollars. Next, let's calculate the cost of the bananas: - Sarah buys 6 bananas. - Each banana costs $1. - Total cost for bananas = $(6 \times 1 = 6)$ dollars. Then, let's calculate the cost of the oranges: - Sarah buys 2 oranges. - Each orange costs $3. - Total cost for oranges = $(2 \times 3 = 6)$ dollars. Now, we sum the costs of all the fruits to find the total amount spent: - Total cost = Cost of apples + Cost of bananas + Cost of oranges - Total cost = 8 + 6 + 6 - Total cost = 20 dollars Therefore, Sarah spends a total of 20 dollars. **Time: 3.5s** |

Table 22: Example – Qualitative comparison for different $\gamma$, LLaDA-1.5

**Prompt:** *Tom has 24 marbles. He gives 1/3 of them to his sister and 1/4 of the remaining marbles to his brother. How many marbles does Tom have left?*

| $\gamma = 0.5$ | $\gamma = 0.7$ | $\gamma = 0.95$ |
|---|---|---|
| To determine how many marbles Tom has left, we will follow these steps: 1. Calculate the number of marbles Tom gives to his sister. 2. Subtract the marbles given to his sister from the original number of marbles. 3. Calculate the number of marbles Tom gives to his brother. 4. Subtract the marbles given to his brother from the remaining marbles. Let's start with the first step: **Step 1: Calculate the number of marbles Tom gives to his sister.** Tom gives $\frac{1}{3}$ of his 24 marbles to his sister. $\frac{1}{3} \times 24 = 8$ So, Tom gives 8 marbles to his sister. **Step 2: Subtract the marbles given to his sister from the original number of marbles.** $24 - 8 = 16$ So, after giving 8 marbles to his sister, Tom has 16 marbles left. **Step 3: Calculate the number of marbles Tom gives to his brother.** Tom gives $\frac{1}{4}$ of the remaining 16 marbles to his brother. $\frac{1}{4} \times 16 = 4$ So, Tom gives 4 marbles to his brother. **Step 4: Subtract the marbles given to his brother from the remaining marbles.** $16 - 4 = 12$ So, after giving 4 marbles to his brother, Tom has 12 marbles left. Therefore, the number of marbles Tom has left is $\boxed{12}$. **Time: 3.3s** | To determine how many marbles Tom has left, we will follow these steps: 1. Calculate the number of marbles Tom gives to his sister. 2. Subtract the marbles given to his sister from the original number of marbles. 3. Calculate the number of marbles Tom gives to his brother. 4. Subtract the marbles given to his brother from the remaining marbles. Let's start with the first step: **Step 1: Calculate the number of marbles Tom gives to his sister.** Tom gives $\frac{1}{3}$ of his 24 marbles to his sister. Marbles given to sister = $\frac{1}{3} \times 24 = 8$ **Step 2: Subtract the marbles given to his sister from the original number of marbles.** Remaining marbles after giving to sister = $24 - 8 = 16$ **Step 3: Calculate the number of marbles Tom gives to his brother.** Tom gives $\frac{1}{4}$ of the remaining 16 marbles to his brother. Marbles given to brother = $\frac{1}{4} \times 16 = 4$ **Step 4: Subtract the marbles given to his brother from the remaining marbles.** Remaining marbles after giving to brother = $16 - 4 = 12$ Therefore, the number of marbles Tom has left is $\boxed{12}$. **Time: 3.1s** | To determine how many marbles Tom has left, we will follow these steps: 1. Calculate the number of marbles Tom gives to his sister. 2. Subtract the marbles given to his sister from the total number of marbles. 3. Calculate the number of marbles Tom gives to his brother. 4. Subtract the marbles given to his brother from the remaining marbles. **Step 1: Calculate the number of marbles Tom gives to his sister.** Tom gives $\frac{1}{3}$ of his 24 marbles to his sister. Marbles given to sister = $\frac{1}{3} \times 24 = 8$ **Step 2: Subtract the marbles given to his sister from the total number of marbles.** Remaining marbles after giving to sister = $24 - 8 = 16$ **Step 3: Calculate the number of marbles Tom gives to his brother.** Tom gives $\frac{1}{4}$ of the remaining 16 marbles to his brother. Marbles given to brother = $\frac{1}{4} \times 16 = 4$ **Step 4: Subtract the marbles given to his brother from the remaining marbles.** Remaining marbles after giving to brother = 16 - 4 = 12 Therefore, the number of marbles Tom has left is $\boxed{12}$. **Time: 3.2s** |

Table 23: Example – Qualitative comparison of responses under different $\gamma$.

**Prompt:** *Points $A$ and $B$ are on parabola $y = 4x^2 + 7x - 1$, and the origin is the midpoint of $\overline{AB}$.*
*Find the square of the length of $\overline{AB}$.*

| $\gamma = 0.5$ | $\gamma = 0.7$ | $\gamma = 0.9$ |
|---|---|---|
| Let $A = (x_1, y_1)$ and $B = (x_2, y_2)$. Since the origin is the midpoint of $\overline{AB}$, we have: | Let $A = (x_1, y_1)$ and $B = (x_2, y_2)$. Since the origin is the midpoint of $\overline{AB}$, we have: | Let $A = (x_1, y_1)$ and $B = (x_2, y_2)$. Since the origin is the midpoint of $\overline{AB}$, we have: |

$\gamma = 0.5$:

Let $A = (x_1, y_1)$ and $B = (x_2, y_2)$. Since the origin is the midpoint of $\overline{AB}$, we have:

$$\left(\frac{x_1 + x_2}{2}, \frac{y_1 + y_2}{2}\right) = (0, 0).$$

This implies:

$$x_1 + x_2 = 0 \quad \text{and} \quad y_1 + y_2 = 0.$$

Since $A$ and $B$ lie on the parabola $y = 4x^2 + 7x - 1$, we have:

$$y_1 + y_2 = (4x_1^2 + 7x_1 - 1) + (4x_2^2 + 7x_2 - 1)$$

Now, using the equations $x_1 + x_2 = 0$ and $y_1 + y_2 = 0$, we get:

$$4x_1^2 + 7x_1 - 1 + 4x_2^2 + 7x_2 - 1 = 0.$$

This simplifies to:

$$4x_1^2 + 7x_1 + 4x_2^2 + 7x_2 - 2 = 0.$$

Thus, the square of the length of $\overline{AB}$ is:

$$(x_1 - x_2)^2 + (y_1 - y_2)^2 = 0^2 + 0^2 = 0.$$

Therefore, the square of the length of $\overline{AB}$ is:

$$\boxed{0}.$$

**Time: 8.1s**

$\gamma = 0.7$:

Let $A = (x_1, y_1)$ and $B = (x_2, y_2)$. Since the origin is the midpoint of $\overline{AB}$, we have:

$$\left(\frac{x_1 + x_2}{2}, \frac{y_1 + y_2}{2}\right) = (0, 0).$$

This implies:

$$x_1 + x_2 = 0 \quad \text{and} \quad y_1 + y_2 = 0.$$

Since $A$ and $B$ lie on the parabola $y = 4x^2 + 7x - 1$, we have:

$$y_1 = 4x_1^2 + 7x_1 - 1$$

and

$$y_2 = 4x_2^2 + 7x_2 - 1.$$

From $x_1 + x_2 = 0$, we get $x_1 = -x_2$. Substituting $x_1 = -x_2$ into the equations for $y_1$ and $y_2$:

$$y_1 = 4x_2^2 + 7x_2 - 1$$

and

$$y_2 = 4x_2^2 + 7x_2 - 1.$$

Since $y_1 + y_2 = 0$, we have:

$$y_1 = -y_2.$$

Thus,

$$y_1 = -(4x_2^2 + 7x_2 - 1) = -4x_2^2 - 7x_2 + 1$$

Similarly,

$$y_2 = 4x_2^2 + 7x_2 - 1.$$

Therefore,

$$y_1 = -y_2 =$$

$$-(4x_2^2 + 7x_2 - 1) = -4x_2^2 - 7x_2 + 1.$$

Thus, the square of the length of $\overline{AB}$ is given by:

$$\boxed{2}.$$

**Time: 9.3s**

$\gamma = 0.9$:

Let $A = (x_1, y_1)$ and $B = (x_2, y_2)$. Since the origin is the midpoint of $\overline{AB}$, we have:

$$\left(\frac{x_1 + x_2}{2}, \frac{y_1 + y_2}{2}\right) = (0, 0).$$

This implies:

$$x_1 + x_2 = 0 \quad \text{and} \quad y_1 + y_2 = 0.$$

Since $A$ and $B$ lie on the parabola $y = 4x^2 + 7x - 1$, we have:

$$y_1 = 4x_1^2 + 7x_1 - 1$$

and

$$y_2 = 4x_2^2 + 7x_2 - 1.$$

Using $y_1 + y_2 = 0$, we get: $y_1 + y_2 = (4x_1^2 + 7x_1 - 1) + (4x_2^2 + 7x_2 - 1) = 0.$ Simplifying, we have:

$$4x_1^2 + 7x_1 + 4x_2^2 + 7x_2 - 2 = 0.$$

Since $x_1 + x_2 = 0$, we get:

$$4x_1^2 + 7x_1 + 4x_2^2 + 7x_2 - 2 = 0.$$

Now, we need to find the square of the length of $\overline{AB}$. The length of $\overline{AB}$ is given by:

$$AB = \sqrt{(x_2 - x_1)^2 + (y_2 - y_1)^2}.$$

Using the midpoint formula and the properties of the parabola, we can simplify the expression. However, since the problem asks for the square of the length, we can directly use the result. Thus, the square of the length of $\overline{AB}$ is:

$$\boxed{50}.$$

**Time: 8.7s (Correct)**

Table 24: Coding Example – Close Elements

**Prompt:**

```python
from typing import List

def has_close_elements(numbers: List[float], threshold: float) -> bool:
    """ Check if in given list of numbers, are any two numbers closer to
        each other than
    given threshold.
    >>> has_close_elements([1.0, 2.0, 3.0], 0.5)
    False
    >>> has_close_elements([1.0, 2.8, 3.0, 4.0, 5.0, 2.0], 0.3)
    True
    """
```

| FAST-DLLM | OURS |
|---|---|
| ```python
sorted_numbers = sorted(numbers)
for i in range(len(sorted_numbers)):
    if abs(sorted_numbers[i] -
        sorted_numbers[i + 1]) <
        threshold:
        return True
return False
``` | ```python
sorted_numbers = sorted(numbers)
for i in range(len(sorted_numbers) -
    1):
    if abs(sorted_numbers[i] -
        sorted_numbers[i + 1]) <
        threshold:
        return True
return False
``` |
| **Fast-DLLM generated incorrect solution** ||

Table 25: Coding Example – Truncate Number

**Prompt:**

```python
def truncate_number(number: float) -> float:
    """ Given a positive floating point number, it can be decomposed
        into
    and integer part (largest integer smaller than given number) and
        decimals
    (leftover part always smaller than 1).

    Return the decimal part of the number.
    >>> truncate_number(3.5)
    0.5
    """
```

| FAST-DLLM | OURS |
|---|---|
| ```python
if number < 0:
    raise ValueError("The number must
        be positive.")
integer_part = int(number)
decimal_part = number - integer_part
return decimal_part
``` | ```python
if number < 0:
    raise ValueError("The number must
        be positive")
integer_part = int(number)
decimal_part = number - integer_part
return decimal_part
``` |
| **Both solutions are functionally equivalent with minor differences** ||

