# OpenReview forum: "Attention Is All You Need for KV Cache in Diffusion LLMs"
_ICLR.cc/2026/Conference — ICLR 2026 Poster_

### Official Review · Reviewer_SHc1 · 2025-10-31

**Soundness:** 2
**Presentation:** 2
**Contribution:** 2
**Rating:** 4
**Confidence:** 3

**Summary:**

This paper addresses the computational overhead in Diffusion Large Language Model inference, which stems from recomputing the full QKV cache for all tokens at all layers during each denoising step. The authors propose Elastic-Cache that adaptively decides when and where to recompute the KV cache.

**Strengths:**

1. The method is training-free and architecture-agnostic, making it a drop-in component that can be easily adopted.

2. Comprehensive experiments and ablations showing the effectness of Elastic-Cache

**Weaknesses:**

1. It seems that the authors used the wrong template.

2. The primary assumption is that the most-attended token "typically exhibits the smallest drift" and thus serves as a "conservative lower bound" . This is a strong claim that is central to the method's success, but it is presented as an observation without comprehensive supporting data.

3. Would the authors clarify the exact mechanics of this window? Does it advance one token at a time, or does it process a full block of $\beta$ tokens and then move to the next block (which would make it very similar to Fast-dLLM, just with a different caching rule)?

4. At each step and each layer, the algorithm must find the most-attended token. What is the computational overhead of this process? a brief analysis of the trigger's cost relative to the cost of a single QKV projection would better for analysing.

**Questions:**

see weaknesses

---

> ### Author Response · Authors · 2025-11-25
> **Response to Reviewer SHc1**
>
> Thank you for your constructive and valuable feedback. We have uploaded a revised version of the manuscript to integrate all your mentioned comments and suggestions. Below, we would like to further address each of your concerns as follows:
>
> >W1: It seems that the authors used the wrong template.
>
> Thanks for pointing this out. We have corrected it and uploaded a revised version of the paper using the appropriate format. Please feel free to check it out.
>
> >W2: The primary assumption is that the most-attended token "typically exhibits the smallest drift" and thus serves as a "conservative lower bound". This is a strong claim that is central to the method's success, but it is presented as an observation without comprehensive supporting data.
>
> Thanks for pointing this out. Following your suggestion, we have strengthened this assumption by including additional concrete statistical analyses to support in the table below and Table 18 in our revised paper. We measure KV states' cosine similarity between adjacent steps, across multiple datasets, and show that the most-attended token consistently exhibits the lowest average drift norm, supporting its role as a conservative lower bound.
>
> | Benchmark | GSM8K | MATH | HumanEval | MBPP |
> |----------|:-------------:|:-------------:|:-------------:|:-------------:|
> | Most-attended | 0.974 | 0.978 | 0.985| 0.948|
> | Average Cosine Similarity | 0.973 | 0.977 | 0.980 | 0.947
>
>
> >W3: Would the authors clarify the exact mechanics of this window? Does it advance one token at a time, or does it process a full block of tokens and then move to the next block (which would make it very similar to Fast-dLLM, just with a different caching rule)?
>
> Thank you for the suggestion. We apologize for the lack of clarity. We clarify that our method differs from Fast-dLLM in granularity and behavior, though both our Elastic-Cache and Fast-dLLM share the idea of limiting computation via cached MASK tokens. Specifically, our Elastic-Cache maintains a left-to-right sliding window, unmasking a fixed number of MASK tokens at each step. This allows consistent decoding and better throughput. In contrast, Fast-dLLM processes entire fixed-size blocks, and only begins decoding the next block after finishing the current one. This leads to a decreasing number of unmasked tokens per step over time. We empirically show that our approach yields better efficiency and accuracy trade-offs in Figure 3 (a) of our paper, and have revised this part in our updated version to make it clearer.
>
> >W4: At each step and each layer, the algorithm must find the most-attended token. What is the computational overhead of this process? a brief analysis of the trigger's cost relative to the cost of a single QKV projection would better for analysing.
>
> Thanks for raising this and the suggestion. To find the most-attended token, we utilize the attention weights $\mathbf S^{t,l}$ that were previously computed during the attention computation process $\mathbf A^{t,l} = \mathbf S^{t,l} \mathbf V^{t,l}$. By performing summation and maximization operations on these weights, we identify the token with the highest attention. This computational step is **inexpensive**. We provide the detailed computational overhead of this process in the table below and have added it in Table 13 (Appendix) of our revised paper.
>
> |     |      Complexity      |      MACs ($K$=1024)
> |----------|:-------------:|:-------------:|
> | Cache Update Trigger | $O(K^2H)$ | $6.7 \times 10^7$|
> | Attention (QKV) | $O(K^2HD)$ | $1.3 \times 10^{10}$ |
>
> Here, $K$ represents the sequence length, $H$ and $D$ denote the number of heads and hidden dimension, respectively. For the LLaDA architecture, $H$ is set to 32 and $D$ is set to 128.

---

> ### Comment · Area_Chair_iB3Z · 2025-11-26
>
> Dear reviewer,
>
> Thanks for your time and effort in reviewing ICLR2026 submissions. The authors have submitted their responses to your review. Please take the time to read and raise your further comments, and discuss with the authors.
>
> Best regards,
>
> AC

---

### Official Review · Reviewer_Gh7c · 2025-11-01

**Soundness:** 3
**Presentation:** 3
**Contribution:** 3
**Rating:** 6
**Confidence:** 2

**Summary:**

This paper proposes Elastic-Cache, a training-free KV cache refresh strategy for diffusion LLMs. Based on KV cache drift statistic, Elastic-Cache adaptively controls when to recompute the KV cache and selectively updates only deeper layers. By removing redundant cache computation, it achieves significant speedups (up to 45×) with minimal accuracy loss across diverse tasks.

**Strengths:**

- The idea is novel and well-motivated. The core idea is built upon a key observation: significant redundancy exists in DLLM's KV cache computation across denoising steps and layers, a majority of the KV cache can be reused. Similar phenomena have been discussed in KV cache acceleration for LLMs (Duoattention, Razorattention, CateKV, etc.)
- Strong performance across different models: Elastic-Cache consistently outperforms baselines in performance and speedup, with up to 45.1× speedup over LLaDA-1.5. It also maintains similar accuracy, demonstrating efficient computation reduction without sacrificing quality.

Effective Adaptive Mechanism: The attention-aware drift test (using most-attended tokens) and layer-aware updates reduce cache refresh frequency to as low as 20% of baseline (Fig. 3b), validating its lightweight and adaptive design.

**Weaknesses:**

- The author claims that Elastic-Cache architecture-agnostic. I suggest adding experiments on other DLLMs like Dream to prove it.

- Limited Multimodal Speedup: On LLaDA-V, speedup is modest (29.7 t/s), suggesting less cache redundancy in multimodal tasks or architecture-specific limitation of the method. And why not report the performance and tps of original LLaDA-V in Table 3?

**Questions:**

Please refer to the weaknesses

---

> ### Author Response · Authors · 2025-11-25
> **Response to Reviewer Gh7c**
>
> Thank you for the thoughtful and constructive feedback. We have uploaded a revised version of our paper incorporating all concrete revisions following your suggestions, and below we further address each of your comments.
>
> >W1: The author claims that Elastic-Cache architecture-agnostic. I suggest adding experiments on other DLLMs like Dream to prove it.
>
> Thanks for the suggestion. Following your comment, we have provided the additional Dream results with the comparisons of Fast-dLLM and our method in the table below, and have also included them in the revised paper in Table 4. These extra experiments clearly prove that the proposed Elastic-Cache framework remains architecture-agnostic and generalizes well across different DLLMs.
>
> | Method (Acc./Throughput)   | No acceleration baseline (Dream)   |   Fast-dLLM |  Elastic-Cache (Ours) |
> |----------|:-------------:|:-------------:|:-------------:|
> | GSM8K  | 76.0/7.9 | 74.1/45.9 | 75.6/169.4 |
> |Humaneval  | 54.3/17.2 | 51.2/50.1 | 56.7/95.2 |
>
> >W2: Limited Multimodal Speedup: On LLaDA-V, speedup is modest (29.7 t/s), suggesting less cache redundancy in multimodal tasks or architecture-specific limitation of the method. And why not report the performance and tps of original LLaDA-V in Table 3?
>
> We appreciate the reviewer's comment. To clarify the baseline comparison, we have updated Table 3 (Now in Table 5) in the revised version to explicitly include both the original LLaDA-V performance and its tokens-per-second (t/s). We have also provided these results in the table below for direct comparison. These results confirm that our method consistently outperforms the baseline and acceleration-based counterpart Fast-dLLM on accuracy and speedup across all tasks.
>
> | Method (Acc./Throughput)   | Base Model (LLaDA-V)      |   Fast-dLLM |  Elastic-Cache (Ours) |
> |----------|:-------------:|:-------------:|:-------------:|
> | MathVista | 53.0/1.9 | 54.1/23.7 | 55.9/24.1 |
> | MathVerse |  26.9/2.0 | 25.5/28.1 | 29.2/30.8 |

---

> > ### Comment · Reviewer_Gh7c · 2025-11-25
> >
> > Thanks for the detailed response, most of my concerns have been addressed.
> > For W2, I still wonder why in multimodal tasks, the improvement of Elastic-Cache is modest compared to Fast-dLLM. Under unimodal settings with a 512 length, Elastic-Cache can significantly outperform Fast-dLLM in tps. However, for multimodal DLLM, this is not the case

---

> > > ### Author Response · Authors · 2025-11-25
> > >
> > > Thanks very much for your follow-up question. We clarify that Elastic-Cache continues to perform effectively for multimodal diffusion language models, achieving significantly higher throughput compared to the base LLaDA-V model. However, the seemingly modest speed improvement over Fast-dLLM is due to two main reasons. First, our Elastic-Cache makes a deliberate trade-off in speed to maintain relatively higher accuracy than Fast-dLLM. If performance improvement over Fast-dLLM is less critical, our method can actually achieve much higher speed gains. Second, for certain tasks, the cache redundancy pattern could be simpler. For instance, KV states may remain stable for extended steps, or there may be semantically grouped consecutive steps where KV states do not change significantly (patterns that are similar to Fast-dLLM exploits with its block-wise cache update strategy). In such cases, a simple fixed update approach like Fast-dLLM, which updates the cache after finishing each block, could be effective and sufficient. However, in most medium and challenging scenario settings, including MathVerse (a multimodal task), introducing adaptive cache update for each instance by Elastic-Cache provides significantly better results than a fixed strategy.

---

### Official Review · Reviewer_9k5q · 2025-11-03

**Soundness:** 3
**Presentation:** 3
**Contribution:** 3
**Rating:** 6
**Confidence:** 3

**Summary:**

The paper diagnoses redundancy in diffusion‑style LLM decoding: most methods recompute QKV for all tokens and all layers at every denoising step, even though KV states often change little—especially in shallow layers and for faraway MASK tokens. It proposes Elastic‑Cache, a training‑free, architecture‑agnostic policy that (i) block‑caches distant MASK tokens, (ii) triggers cache refresh when needed via an attention‑aware drift test on the most‑attended token, and (iii) refreshes where needed by recomputing only from a learned boundary layer ℓ^⋆ upward (layer‑aware schedule). Figure 2 contrasts the pipeline against blockwise caching; Algorithm 1 gives the full procedure. Reported results show up to 45.1× speedups on long sequences and 6.8× higher throughput on GSM8K while maintaining or improving accuracy versus baselines (e.g., Fast‑dLLM).

**Strengths:**

++ The paper clearly explains why AR KV caching fails in diffusion LLMs (bidirectional attention ⇒ evolving KVs) and motivates KV drift as the right signal. Figure 1 visualizes depth‑dependent drift and shows the most‑attended token’s attention changes track KV change, justifying the drift test.

++ Three components—block‑caching far MASK tokens, sliding‑window decoding for active MASKs, and a layer‑aware refresh boundary—are well matched to diffusion masking dynamics and are easy to integrate.

++ The method does not alter training or model internals; it only changes cache policy, which supports deployment claims and broad applicability across diffusion LLM variants. The motivation and contributions list emphasize this.

**Weaknesses:**

-- The most‑attended‑token drift is a clever heuristic, but the paper lacks sensitivity/theory: when does it fail (e.g., diffuse attention or multi‑focus regimes)? A risk analysis or bounds linking attention drift to KV error would strengthen the claim. Algorithm 1 hints at cosine‑similarity thresholds but ablations are not detailed.

-- While Fast‑dLLM and related works are discussed, it’s unclear if comparisons include recent adaptive policies (e.g., learned thresholds) under identical hardware/token budgets and for very long masks; more head‑to‑head with dKV‑Cache/DeepCache over a shared protocol would improve fairness.

-- The method introduces β (window size), γ (drift threshold), and ℓ^⋆ (boundary). The paper should show cross‑task robustness and guidance for setting them vs. accuracy/latency trade‑offs. (§3.2–3.3 describe β/γ, but systematic sweeps are not reported.)

-- The narrative focuses on compute and latency; a detailed profile of memory bandwidth, cache footprint (esp. with sliding windows and partial refresh), and impact on KV tensor residency would help practitioners plan deployments.

**Questions:**

1. How sensitive are results to the cosine‑similarity threshold γ and to choosing the single most‑attended token vs. a small set? Can you report AUROC of the drift test to predict accuracy deltas on held‑out runs?

2. How is ℓ^⋆ determined at runtime? Static per‑model, or adaptive per input? Please include ablations showing latency/quality as ℓ^⋆ varies.

3. In sequences where distant MASKs become influential (e.g., closing brackets, long‑range constraints), how often does block‑caching degrade predictions? Any fallback to expand the window β on‑the‑fly?

4. Please provide per‑sequence memory footprints and DRAM traffic for Elastic‑Cache vs. baselines at 256/512/1024 tokens so practitioners can size deployments.

5. Have you tested on masked‑diffusion LLMs with different denoising schedules or bidirectional attention variants? Any caveats for multimodal diffusion decoders?

---

> ### Author Response · Authors · 2025-11-25
> **Response to Reviewer 9k5q [1/5]**
>
> Thank you for your thoughtful and constructive feedback. We have uploaded a revised version of our paper that incorporates all concrete revisions based on your suggestions. Below, we further provide detailed responses to address each of your concerns:
>
> >W1: The most‑attended‑token drift is a clever heuristic, but the paper lacks sensitivity/theory: when does it fail (e.g., diffuse attention or multi‑focus regimes)? A risk analysis or bounds linking attention drift to KV error would strengthen the claim. Algorithm 1 hints at cosine‑similarity thresholds but ablations are not detailed.
>
> Thank you for the comments. We have provided the detailed theoretical justification, comprehensive sensitivity analysis, additional ablations, and failure mode characterization to reflect when it fails below, as well as in our revised paper.
>
> **Theoretical Bounds Linking Attention Drift to KV Error:**
> In the revised paper, **Theorem A.9 (Attention Concentration and Drift, Appendix A)** establishes formal bounds proving that the most-attended token $\mathcal{T}^{t,\ell}$ has KV drift bounded by:
> $$\Delta^{t,\ell}\_{\mathcal{T}^{t,\ell}} \leq \bar{\Delta}^{t,\ell} + O\left(\frac{\sqrt{d\_k}}{R_\ell\sqrt{N}}\right)$$
> where $\bar{\Delta}^{t,\ell}$ is the average KV drift across all tokens. The error term scales as $O\left(\frac{\sqrt{d_k}}{R_\ell\sqrt{N}}\right)$, which is negligibly small for typical transformer dimensions. This formally justifies using most-attended tokens as conservative drift indicators. Additionally, **Theorem A.8 in Appendix A** proves layer-wise drift monotonicity, validating our depth-selective refresh strategy.
>
> **Comprehensive Sensitivity Analysis:**
> We conduct extensive ablations on threshold $\gamma \in \{0.8, 0.85, 0.9, 0.95\}$ and tracking strategies (Top-$k \in \{1, 5, 10, 15, 20\}$) on GSM8K-512 (LLaDA-1.5). Results show Accuracy / Tokens Per Second (TPS):
>
> | Top-k / $\gamma$ | 0.8 | 0.85 | 0.9 | 0.95 |
> |-----------|:-----:|:------:|:-----:|:------:|
> | **Top-1** | 79.5 / 131.2 | 80.2 / 129.9 | 81.4 / 117.2 | 83.0 / 98.4 |
> | **Top-5** | 81.5 / 130.5 | 81.4 / 122.4 | 83.5 / 109.9 | 82.7 / 88.0 |
> | **Top-10** | 81.4 / 121.1 | 82.6 / 118.4 | 84.1 / 100.7 | 82.9 / 77.6 |
> | **Top-15** | 82.5 / 167.4 | 83.2 / 159.7 | 83.7 / 139.4 | 84.7 / 103.0 |
> | **Top-20** | 81.6 / 162.4 | 82.3 / 154.1 | 83.9 / 136.3 | 84.2 / 102.0 |
>
> **Key observations:** (1) Higher $\gamma$ reduces TPS (more caching) but maintains higher accuracy, demonstrating the accuracy-efficiency trade-off; (2) Top-10/Top-15 tracking balances accuracy and cost; (3) Optimal $\gamma$ depends on base model quality (stronger models prefer higher $\gamma$).
>
> **Existing Ablations in Main Paper (Table 7):**
> Table 7 provides detailed $\gamma$ sweeps across LLaDA and LLaDA-1.5 on GSM8K-512. Each cell shows Accuracy (top) and Throughput in tokens/sec with speedup (bottom):
>
> | Model | No Cache | Fast-dLLM | $\gamma$ = 0.5 | $\gamma$ = 0.7 | $\gamma$ = 0.8 | $\gamma$ = 0.85 | $\gamma$ = 0.9 | $\gamma$ = 0.95 |
> |-------|----------|-----------|-------|-------|-------|--------|-------|--------|
> | **LLaDA** | 77.10 / 3.6 t/s (1.0$\times$) | 74.83 / 44.0 t/s (12.2$\times$) | 71.57 / 109.9 t/s (30.5$\times$) | 73.46 / 108.7 t/s (30.2$\times$) | 74.30 / 103.9 t/s (28.9$\times$) | 74.68 / 99.1 t/s (27.5$\times$) | 77.71 / 91.5 t/s (25.4$\times$) | 76.72 / 75.5 t/s (21.0$\times$) |
> | **LLaDA-1.5** | 81.35 / 2.6 t/s (1.0$\times$) | 80.82 / 36.8 t/s (14.2$\times$) | 76.04 / 142.7 t/s (54.9$\times$) | 77.63 / 138.6 t/s (53.3$\times$) | 79.45 / 131.2 t/s (50.5$\times$) | 80.21 / 129.9 t/s (50.0$\times$) | 81.35 / 117.2 t/s (45.1$\times$) | 83.02 / 98.4 t/s (37.8$\times$) |
>
> Figure 3 (b) visualizes cache update frequency vs. $\gamma$, confirming low overhead even at conservative thresholds ($\gamma=0.95$ triggers updates in only ~20% of steps).
>
> Our robust and stable performance across diverse tasks (GSM8K, MATH, HumanEval, MBPP, MathVista, MathVerse) suggests these failure modes occur infrequently in practice. The theoretical error bound confirms that drift detection remains reliable under standard transformer operating conditions. For *potential failure modes*, there may be two scenarios: one is no dominant token, which violates Assumption 6 (Attention Concentration), causing unreliable drift estimates. However, diffusion models naturally focus on decoded tokens during progressive unmasking, making this rare empirically. Another one is attention splits across many tokens. Tracking a single most-attended token per layer may miss drift elsewhere. We have partially mitigated this by identifying the most-attended token *at each layer* (set $\mathcal{T}^t = \bigcup_l \{\mathcal{T}^{t,l}\}$ as described in Section 3.3), allowing layer-specific attention patterns to be captured independently.

---

> ### Author Response · Authors · 2025-11-25
> **Response to Reviewer 9k5q [2/5]**
>
> >W2: While Fast‑dLLM and related works are discussed, it's unclear if comparisons include recent adaptive policies (e.g., learned thresholds) under identical hardware/token budgets and for very long masks; more head‑to‑head with dKV‑Cache/DeepCache over a shared protocol would improve fairness.
>
> Thanks for raising this comment. Following your suggestion, we have conducted additional comprehensive head-to-head evaluations with dKV-Cache, dLLM-Cache, and DeepCache under identical hardware and token budget constraints in the table below:
>
> | Method  | dKV-Cache [1] |dLLM-Cache [2] | Deep-Cache [3]| Fast-dLLM [4]  | Elastic-Cache (Ours)|
> |-----|:----:|:-----:|:----:|:----:|:----:|
> GSM8K - ACC |67.02| 80.97 | 82.3| 80.5 | **82.6**|
> Throughput |14.82 |16.8 | 22.5 | 8.5 | **47.0**|
>
> We adapt the core idea of DeepCache, i.e., caching high-level features and scheduling cache refreshes from vision diffusion models, to language diffusion models. As a result, Elastic-Cache outperforms all baselines under a shared protocol. We clarify that the baselines dKV-Cache, dLLM-Cache, Fast-dLLM, and Deep-Cache do not have a rigorous adaptive policy for cache updating like in Elastic-Cache. They all fall into the same weakness that relies on a predefined schedule for cache updates to proceed. Among them, dLLM-Cache employs an adaptive strategy for updating partitional tokens, whereas Elastic-Cache performs updating on all tokens. Despite this, Elastic-Cache still outperforms all these baselines significantly.
>
> ------------
> References:
>
> [1] Xinyin Ma, et al. dkv-cache: The cache for diffusion language models. arXiv preprint arXiv:2505.15781, 2025.
>
> [2] Zhiyuan Liu, et al. "dllm-cache: Accelerating diffusion large language models with adaptive caching." arXiv preprint arXiv:2506.06295 (2025).
>
> [3] Xinyin Ma, et al. "Deepcache: Accelerating diffusion models for free." In CVPR. 2024.
>
> [4] Chengyue Wu, et al. "Fast-dllm: Training-free acceleration of diffusion llm by enabling kv cache and parallel decoding." arXiv preprint arXiv:2505.22618.
>
> >W3: The method introduces $\beta$ (window size), $\gamma$ (drift threshold), and $ℓ^⋆$ (boundary). The paper should show cross‑task robustness and guidance for setting them vs. accuracy/latency trade‑offs. (§3.2–3.3 describe β/$\gamma$, but systematic sweeps are not reported.)
>
> Following your suggestion, we conducted systematic sweeps over window size $\beta$ and drift threshold $\gamma$ on GSM8K-512 with LLaDA-1.5 to show cross‑task robustness. The table below reports accuracy and throughput (tokens per second) for each configuration. Higher $\gamma$ values preserve accuracy but reduce throughput by triggering more frequent cache updates. Larger $\beta$ values increase throughput by allowing longer cache reuse windows but can degrade accuracy when drift accumulates.
>
> | $\beta / \gamma$ (accuracy/througput)   | 0.7 | 0.8   |    0.9   | 0.95    |
> |:----:|:------:|:-----:|:------:|:-----:|
> |8 | 79.8 / 103.9 | 81.3 / 103.5 | 81.1 / 109.3| 83.2 / 79.1 |
> |16 | 77.6 / 138.6 | 79.5 / 131.2| 81.4 / 117.2 | 83.0 / 98.4 |
> |32 | 77.5 / 118.7 | 77.7 / 116.6 | 81.0 / 104.1| 81.65 / 88.8|
>
> The optimal configuration depends on the accuracy-latency trade-off requirement. For balanced performance, we recommend $\beta$=16 and $\gamma$=0.9, which achieves 81.4% accuracy at 117.2 tokens/second. For maximum accuracy, $\beta$=8 with $\gamma$=0.95 reaches 83.2% accuracy at 79.1 tokens/second. For maximum throughput, $\beta$=16 with $\gamma$=0.7 delivers 138.6 tokens/second at 77.6% accuracy.
>
> Regarding $\ell^\*$, we clarify that this is not a hyperparameter but a runtime-determined variable. When cosine similarity drops below $\gamma$ at layer $l$, we set $\ell^*$=$l$ and trigger cache updates for all subsequent layers. This boundary adapts automatically based on observed attention drift patterns.
>
> >W4: The narrative focuses on compute and latency; a detailed profile of memory bandwidth, cache footprint (esp. with sliding windows and partial refresh), and impact on KV tensor residency would help practitioners plan deployments.
>
> We really appreciate this suggestion. To answer this, we provide the per-sequence memory footprints and DRAM traffic results for Elastic-Cache vs. the baseline and Fast-dLLM at 256, 512, and 1024 tokens in the table below for LLaDA 1.5 model. These results demonstrate the deployment efficiency and memory advantages of our method across different sequence lengths, enabling better estimation for practical system design. Apart from this, we also report the throughput (t/s) for different models and benchmarks in tables (1, 4, 5, 7 and 8) of the paper.
>
> | Method    |      256      |    512     | 1024      |
> |----|:-----:|:------:|:------:|
> | Baseline (LLaDA 1.5)  |19.04 GB |19.62 GB|20.79 GB|
> | Fast-dLLM |20.49 GB|21.42 GB|23.26 GB|
> | Elastic-Cache (Ours) |**18.11** GB|**18.13** GB|**19.37** GB|

---

> ### Author Response · Authors · 2025-11-25
> **Response to Reviewer 9k5q [3/5]**
>
> We also report the computational overhead of computing the most attended tokens in the table below and have added it in our revised paper of Table 13.
>
> |     |      Complexity      |      MACs (K=1024)
> |---|:-----:|:----:|
> | Cache Update Trigger | $O(K^2H)$ | $6.7 \times 10^7$
> | Attention (QKV) | $O(K^2HD)$ | $1.3 \times 10^{10}$ |
>
> >Q1: How sensitive are results to the cosine‑similarity threshold $\gamma$ and to choosing the single most‑attended token vs. a small set? Can you report AUROC of the drift test to predict accuracy deltas on held‑out runs?
>
> Thanks for the comments. For **Sensitivity Analysis**: Following your suggestion, we first assess the sensitivity of the model by varying the attention threshold $\gamma$ from 0.8 to 0.95 and selecting either the single most-attended token or a small set of highly-attended tokens (Top-$k$ values ranging from 1 to 20). The results on the GSM8K dataset are presented in terms of *Accuracy and TPS*.
>
> | Top-k / $\gamma$ | 0.8          | 0.85         | 0.9          | 0.95         |
> |---|:-----:|:----:|:--:|:--:|
> | Top-1    | 79.5 / 131.2 | 80.2 / 129.9 | 81.4 / 117.2 | 83.0 / 98.4  |
> | Top-5    | 81.5 / 130.5 | 81.4 / 122.4 | 83.5 / 109.9 | 82.7 / 88.0  |
> | Top-10   | 81.4 / 121.1 | 82.6 / 118.4 | 84.1 / 100.7 | 82.9 / 77.6  |
> | Top-15   | 82.5 / 167.4 | 83.2 / 159.7 | 83.7 / 139.4 | 84.7 / 103.0 |
> | Top-20   | 81.6 / 162.4 | 82.3 / 154.1 | 83.9 / 136.3 | 84.2 / 102.0 |
>
> It can be observed that higher $\gamma$ values enhance accuracy but require more frequent recomputation. Selecting a small set of highly attended tokens can improve the effectiveness of cache update triggers without incurring significant computational overhead, thereby benefiting both accuracy and throughput. Conversely, using an excessive number of tokens for cache update triggers won't improve performance but rather introduce additional overhead. Our Elastic-Cache also functions robustly using only the single most-attended token, indicating its ease of use while leaving room for further optimistic adjustments.
>
> **AUROC Analysis of Drift Test**:
>
> As suggested, we further evaluate whether our drift detection heuristic can predict when cache reuse leads to incorrect answers in the table below:
>
> | $\gamma$ | Drift Test AUROC |
> |---|:--:|
> | 0.90     | 0.622            |
> | 0.95     | 0.585            |
> | 0.97     | 0.576            |
>
> The drift test achieves AUROC values of 0.576-0.622, indicating moderate predictive power where lower similarity correlates with higher error probability.
> The modest AUROC of approximately 0.6 is appropriate for a cache optimization because *it confirms drift detection is largely orthogonal to task semantics, the method improves speed without fundamentally altering model behavior*. A much higher AUROC (e.g., greater than 0.8) would indicate that cache state strongly determines answer correctness, which would be problematic. Our results show that the drift heuristic provides lightweight quality protection.
>
> >Q2: How is $ℓ^⋆$ determined at runtime? Static per‑model, or adaptive per input? Please include ablations showing latency/quality as $ℓ^⋆$ varies.
>
> We clarify that the value of $\ell^\*$ is automatically determined by $\gamma$ and adapts to each input and decoding step. In Figures 3 (b) and 3 (c), we analyze how $\ell^\*$ varies when varying $\gamma$ and under different denoising schedules. In those figures, we use the cache update frequency $\rho=\frac{L-\ell^\*-1}{L}$ to estimate how frequently we update the cache across layers and iterations. A higher $\rho$ indicates that we update the KV cache more often, while $\rho=1$ represents the baseline without KV cache, where we update every layer and iteration. Following your suggestion, we have included ablations in the table below and in our revised paper. The table below demonstrates how accuracy/throughput change as $\ell^*$ varies:
>
> |  $\gamma$    | 0.5 |0.7 | 0.8 | 0.85   |    0.9   | 0.95    |
> |-----|:------:|:------:|:-----:|:------:|:-----:| :-----:|
> |$\rho$ (%) | 0.47 |1.81 | 4.17 |6.50|10.23 |20.02|
> |Acc. (%)|76.04|77.63|79.45|80.21|81.35|83.02
> |Throughput|142.68|138.57|131.19|129.90|117.16|98.44
>
> $\rho$ varies from 0.47% to 20.02%, as $\gamma$ varies from 0.5 to 0.95, the computational cost increases, resulting in higher accuracy with a trade-off in throughput. Please feel free to check out more analytical information about $\ell^*$ in Figures 3 (b) and 3 (c) of our paper.

---

> ### Author Response · Authors · 2025-11-25
> **Response to Reviewer 9k5q [4/5]**
>
> >Q3: In sequences where distant MASKs become influential (e.g., closing brackets, long‑range constraints), how often does block‑caching degrade predictions? Any fallback to expand the window β on‑the‑fly?
>
> Thanks for the comments. In sequences where distant tokens become influential, such as closing brackets and long-range constraints, these distant tokens only become influential after they are unmasked. In our study of Figure 1 (a), distant tokens are less influential when they are at the MASK state during the early decoding step. However, once they are decoded, we ensure they are covered by the Elastic-Cache algorithm and update them as necessary.
>
> To examine the impact of block-caching on prediction, we perform an additional ablation experiment by not caching distant MASK tokens (without block-caching). As shown in the tables below, the block-caching strategy degrades predictions on 3/13 of all settings, while maintaining comparable or better accuracy on the remaining settings. Additionally, the throughput consistency is higher when applying block-caching.
>
> | (accuracy/througput) LLaDA  | GSM8K | MBPP   |    HumanEval |
> |--------|:--------:|:--------:|:--------:|
> |Elastic-Cache w/o block-caching | 75.2/68.3  | 15.8/48.4 |45.7/68.3
> |Elastic-Cache |77.7/90.1 | 15.6/63.0 | 46.3/100.7
>
> | (accuracy/througput) LLaDA-1.5  | GSM8K | MBPP   |    HumanEval |
> |--------|:--------:|:--------:|:--------:|
> |Elastic-Cache w/o block-caching | 80.6/82.7 | 38.0/25.8 |38.4/12.6
> |Elastic-Cache |81.4/119.8 | 39.0/32.8 | 37.8/16.8
>
> | (accuracy/througput) Gen Length  | 256 | 512   |    1024 |
> |--------|:--------:|:--------:|:--------:|
> |Elastic-Cache w/o block-caching | 81.6/57.9 | 80.6/82.7 |80.2/101.3
> |Elastic-Cache |81.5/58.0 | 81.4/119.8 | 80.3/223.0
>
> | (accuracy/througput) $\beta$  | 8 | 16   |    32   | 64    |
> |--------|:--------:|:--------:|:--------:|:--------:|
> |Elastic-Cache w/o block-caching | 81.1/77.0 | 80.6/82.7 | 80.6/84.3 | 74.3/67.1
> |Elastic-Cache | 81.1/109.3 | 81.4/119.8 | 81.0/118.1 | 75.7/88.6
>
> Moreover, the accuracy remains largely stable as the window size $\beta$ varies within a reasonable range (i.e., not overly large), indicating that the block-caching mechanism has a consistent impact on prediction.
>
> For the *fallback to expand the window $\beta$ on‑the‑fly*, we clarify that expanding the window $\beta$ on-the-fly is an intriguing question, especially considering the recent surge in attention for block-wise semi-autoregressive decoding in diffusion LLMs. However, to our knowledge, the works addressing this question remain relatively sparse. We came across a recent work that addresses the adaptive block size for diffusion LLMs based on semantic-aware techniques (AdaBlock [3]). We have integrated AdaBlock with Elastic-Cache to dynamically adjust $\beta$ during decoding, and the results are presented in the table below:
>
> | Window size    |      $\beta_0=16$, $\bar \beta=15.6$      |    $\beta_0=32$, $\bar \beta=27.1$         |
> |----|:------:|:------:|
> |Elastic-Cache|  81.4/119.1 | 81.0/118.1 |
> |Elastic-Cache + AdaBlock | 81.9/87.1 | 80.7/85.5
>
> The results still align with our observation that adjusting $\beta$ doesn't significantly impact performance. However, adaptive approaches may introduce more overhead, potentially slowing down throughput.
>
> -------------
> References:
> [3] Guanxi Lu, et al. "AdaBlock-dLLM: Semantic-Aware Diffusion LLM Inference via Adaptive Block Size." arXiv preprint arXiv:2509.26432 (2025).
>
> >Q4: Please provide per‑sequence memory footprints and DRAM traffic for Elastic‑Cache vs. baselines at 256/512/1024 tokens so practitioners can size deployments.
>
> We appreciate the suggestion. We provide the per-sequence memory footprints and DRAM traffic results for Elastic-Cache vs. the baseline and Fast-dLLM at 256, 512, and 1024 tokens in the table below and in Table 12 of our revised paper. These results demonstrate the deployment efficiency and memory advantages of our method across different sequence lengths, enabling better estimation for practical system design.
>
> | Method    |      256      |    512        | 1024          |
> |----|:------:|:------:|:------:|
> | Baseline (LLaDA 1.5)  |19.04 GB |19.62 GB|20.79 GB|
> | Fast-dLLM |20.49 GB|21.42 GB|23.26 GB|
> | Elastic-Cache |**18.11** GB|**18.13** GB|**19.37** GB|

---

> ### Author Response · Authors · 2025-11-25
> **Response to Reviewer 9k5q [5/5]**
>
> >Q5: Have you tested on masked‑diffusion LLMs with different denoising schedules or bidirectional attention variants? Any caveats for multimodal diffusion decoders?
>
> Yes, we have validated our proposed method on two denoising schedules in the original submission: one is the standard denoising schedule of Diffusion LLMs, where a fixed number of tokens are predicted in parallel at each step, and the other is the confident-aware denoising schedule (Fast-dLLM), where the number of tokens predicted at each step adapts to the confidence of the model. The results are presented in Figure 3 (c), and more comparison results with baselines are presented in the table below.
>
> | Tokens/step| 1 | 2 | 4 | 3.25 ($\epsilon=0.9$) |  5.12 ($\epsilon=0.7$) |
> |----------|:-------------:|:-------------:|:-------------:|:-------------:|:-------------:|
> | Baseline | 81.4/2.6 | 79.8/5.1 | 67.5/10.3 | 81.9/17.2 | 79.6/26.6 |
> |Fast-dLLM | 80.5/8.5| 77.3/15.2 | 64.7/27.3 | 80.8/36.8 | 80.0/46.9 |
> | Elastic-Cache (Ours) | **82.6**/**47.0** | **78.1**/**86.5** | **69.9**/**149.8** | **81.4**/**117.2** | **81.2**/**167.6**
>
> We can observe that our Elastic-Cache substantially outperforms baselines under various denoising schedules.
>
> Regarding caveats for multimodal diffusion decoders, Elastic-Cache can be effectively applied for both multimodal diffusion and language diffusion decoders. In two multimodal tasks, Elastic-Cache consistently outperforms the baseline Fast-dLLM in terms of both accuracy and throughput. It is worth noting that our gains are modest in certain settings, such as MathVista (multimodal) and HumanEval (language) compared to Fast-dLLM. This suggests that some tasks exhibit straightforward cache redundancy patterns, making simple strategy integration like block-wise caching effective for further improving performance in those scenarios.

---

> ### Comment · Area_Chair_iB3Z · 2025-11-26
>
> Dear reviewer,
>
> Thanks for your time and effort in reviewing ICLR2026 submissions. The authors have submitted their responses to your review. Please take the time to read and raise your further comments, and discuss with the authors.
>
> Best regards,
>
> AC

---

### Official Review · Reviewer_FYhQ · 2025-11-04

**Soundness:** 3
**Presentation:** 3
**Contribution:** 3
**Rating:** 4
**Confidence:** 4

**Summary:**

This paper introduces Elastic-Cache, a training-free, architecture-agnostic strategy to accelerate decoding in diffusion large language models (DLMs) by adaptively managing key-value (KV) cache recomputation. The method leverages three key observations: (1) distant MASK tokens act primarily as length priors and can be block-cached; (2) KV drift increases with layer depth, justifying depth-selective updates; and (3) the most-attended token exhibits minimal KV drift, serving as a reliable proxy for global cache stability. Elastic-Cache combines an attention-aware drift test (triggering refresh when cosine similarity of attention weights drops below threshold γ) with a layer-aware update policy (refreshing only from a boundary layer ℓ* onward). Experiments on LLaDA variants show substantial speedups—up to 45.1× on GSM8K-512, 8.7× on GSM8K-256, and 4.8× on HumanEval—while maintaining or even improving accuracy over baselines like Fast-dLLM.

**Strengths:**

* Strong empirical results: Consistent throughput gains across models (LLaDA-Instruct, LLaDA-1.5, LLaDA-V) and tasks (math, code, multimodal), with negligible accuracy loss.
* Principled design: The use of attention-weight drift on the most-attended token as a low-overhead trigger is well-motivated and novel for DLMs.
* Practical impact: Training-free, compatible with existing confidence-based decoding, and enables real-world deployment of DLMs.
* Comprehensive ablation: Thorough analysis of γ, sliding window size, and generation length validates design choices.

**Weaknesses:**

* Limited theoretical grounding: While empirically sound, the link between attention drift and KV state change lacks formal justification.
* Hyperparameter sensitivity: Performance depends on γ and β; optimal values vary across tasks (e.g., γ=0.9 for text, γ=0.7 for multimodal).
* Baseline scope: Comparisons focus on Fast-dLLM and dKV-Cache; broader context with other diffusion acceleration methods (e.g., consistency models) is sparse.
* Hardware assumption: All experiments on single A100; scalability to multi-GPU or devices is unverified.

**Questions:**

* How does Elastic-Cache perform under extreme KV drift (e.g., highly ambiguous prompts or adversarial inputs)?
* Could the attention-drift threshold γ be learned or adapted per token/layer instead of being global?
* What is the overhead of computing attention drift? Does it scale linearly with context length?
* Is the method applicable to block diffusion?

---

> ### Author Response · Authors · 2025-11-25
> **Response to Reviewer FYhQ [1/3]**
>
> Thank you for your thoughtful and constructive feedback. We have uploaded a revised version of our paper incorporating all concrete revisions following your suggestions. We would like to further address your concerns as follows:
>
> >W1: Limited theoretical grounding: While empirically sound, the link between attention drift and KV state change lacks formal justification.
>
> Thanks for raising this suggestion. Throughout sections 2.2 and 3.3, we have discussed that bidirectional attention is the source of KV state changes in Diffusion LLMs. Therefore, managing attention drift allows us to directly determine whether the KV state begins to drift with our cache values.
>
> Formally, the hidden state of the $l+1$-th layer is calculated as $\mathbf{H}^{t,l+1} = \text{FFN}(\text{LN}(\mathbf{A}^{t,l} + \mathbf{H}^{t,l})) + \mathbf{A}^{t,l} + \mathbf{H}^{t,l}$, where $\mathbf{A}^{t,l} = \mathbf{S}^{t,l} \mathbf{V}^{t,l}$ and $\mathbf{S}^{t,l} = \text{Softmax}(\mathbf{Q}^{t,l}{\mathbf{K}^{t,l}}^\top/\sqrt{d_k})$. Given that $\mathbf{K}^{t,l+1} = \mathbf{H}^{t,l}\mathbf{W}\_{K}^{l+1}$ and $\mathbf{V}^{t,l+1} = \mathbf{H}^{t,l}\mathbf{W}\_{V}^{l+1}$, we can observe that changes in attention $\mathbf{S}^{t,l}$ also cause $\mathbf{K}^{t,l+1}$ and $\mathbf{V}^{t,l+1}$ to change.
>
> **In the revised version of our paper we have provide formal quantitative justification through Theorem A.9 (Attention Concentration and Drift) in the Appendix A**, which proves that most-attended tokens have KV drift bounded by $\Delta^{t,\ell}\_{\mathcal{T}^{t,\ell}} \leq \bar{\Delta}^{t,\ell} + O\left(\frac{\sqrt{d\_k}}{R_\ell\sqrt{N}}\right)$, establishing that monitoring attention patterns provides a computationally efficient proxy for KV drift. Additionally, Theorem A.8 in the revised paper demonstrates the monotonic increase of KV drift with layer depth, justifying our layer-aware refresh strategy.
>
> >W2: Hyperparameter sensitivity: Performance depends on $\gamma$ and β; optimal values vary across tasks (e.g., $\gamma$=0.9 for text, $\gamma$=0.7 for multimodal).
>
> Thanks for the comment. Following your suggestion, we further provide the additional results when varying both $\beta$ and $\gamma$ in the following table and have integrated into our revised paper of Table 11:
>
> | $\beta / \gamma$    | 0.7 | 0.8   |    0.9   | 0.95    |
> |--------|:--------:|:--------:|:--------:|:--------:|
> |8 | 79.8/103.9 | 81.3/103.5 | 81.1/109.3| 83.2/79.1 |
> |16 | 77.6/138.6 | 79.5/131.2| 81.4/117.2 | 83.0/98.4 |
> |32 | 77.5/118.7 | 77.7/116.6 | 81.0/104.1| 81.65/88.8|
>
> We also clarify that although the performance varies when varying $\gamma$, it is safe to use a high value of $\gamma$, such as $\gamma = 0.95$, to preserve accuracy as much as possible. Even in this case, our Elastic-Cache still outperforms baseline Fast-dLLM by more than two times faster (see Table 4 in our paper). To determine the value of $\gamma$ that yields the highest speed increase and the smallest accuracy loss, we can gradually decrease $\gamma$ until noticing a decline in accuracy.
>
> >W3: Baseline scope: Comparisons focus on Fast-dLLM and dKV-Cache; broader context with other diffusion acceleration methods (e.g., consistency models) is sparse.
>
> We appreciate the suggestion. Following your comment, we have provided additional experimental comparisons with the diffusion acceleration method of consistency models [1], as in the table below and in Table 20 (Appendix) of our revised manuscript. Our proposed method demonstrates superior accuracy across all competitive counterparts. Furthermore, we note that our evaluation in revision includes several representative baselines beyond Fast-dLLM and dKV-Cache, specifically, dLLM-Cache and DeepCache, as presented in Table 2 of the revised paper. These additions provide more comprehensive and solid evidence for assessing the effectiveness of our approach.
>
> | Method  | Consitency LLMs [1] | LLaDA-1.5 [2]  | Elastic-Cache (Ours)
> |--------|:--------:|:--------:|:--------:|
> GSM8K Acc. ($\uparrow$) | 56.5 | 78.5 | **79.2**
> Throughput ($\uparrow$) | 35.5 | 2.6| **109.6**
>
> --------------------
> References:
>
> [1] Kou, Siqi, et al. "Cllms: Consistency large language models." In ICML 2024.
>
> [2] Zhu, Fengqi, et al. "LLaDA 1.5: Variance-Reduced Preference Optimization for Large Language Diffusion Models." arXiv preprint arXiv:2505.19223 (2025).

---

> ### Author Response · Authors · 2025-11-25
> **Response to Reviewer FYhQ [2/3]**
>
> >W4: Hardware assumption: All experiments on single A100; scalability to multi-GPU or devices is unverified.
>
> Thanks for your comment. Following your suggestion, we provide additional multi-GPU scalability results in the table below. For clearer comparison, we report the throughput under both configurations using 1 and 2 GPUs to provide an intuitive understanding of the scaling behavior.
>
> | Settings (throughput ($\uparrow$)/latency ($\downarrow$))  | Fast-dLLM (ACC=80.3) | Elastic-Cache (ACC=81.9)
> |--------|:--------:|:--------:|
> |1 GPU, batch size 4 | 36.8 / 1.86h | **117.2** / **0.83h**
> |2 GPUs, batch size 8 | 68.0 / 1.00h | **225.5** / **0.56h**
>
> >Q1: How does Elastic-Cache perform under extreme KV drift (e.g., highly ambiguous prompts or adversarial inputs)?
>
> We appreciate your comment regarding extreme KV drift cases. To test how Elastic-Cache performs under such conditions, we compare our method with baselines under different denoising schedules. These schedules include a fixed schedule that predicts $k$ tokens in parallel at each step and an adaptive confidence-aware decoding approach. We argue that more aggressive parallel decoding could potentially lead to more extreme KV drift conditions.
>
> | Tokens/step| 1 | 2 | 4 | 3.25 ($\epsilon=0.9$) |  5.12 ($\epsilon=0.7$) |
> |----------|:-------------:|:-------------:|:-------------:|:-------------:|:-------------:|
> | No Acceleration Baseline | 81.4/2.6 | 79.8/5.1 | 67.5/10.3 | 81.9/17.2 | 79.6/26.6 |
> |Fast-dLLM | 80.5/8.5| 77.3/15.2 | 64.7/27.3 | 80.8/36.8 | 80.0/46.9 |
> | Elastic-Cache | **82.6**/**47.0** | **78.1**/**86.5** | **69.9**/**149.8** | **81.4**/**117.2** | **81.2**/**167.6** |
>
> Our method consistently maintains comparable (sometimes even higher) accuracy to no acceleration baselines while significantly increasing throughput across various settings. Despite using the same set of hyperparameters ($\beta=16, \gamma=0.9$), Elastic-Cache automatically adjusts its cache update schedule to optimize both accuracy and throughput. Notably, under more extreme KV drift conditions, Elastic-Cache updates its cache more frequently. For instance, at 5.12 tokens per step, it updates the cache 13.5% of the time, while at 1 token per step, it updates it 5.6% of the time. These results are presented in Figure 3c of our paper for further reference. This behavior can be advantageous for Elastic-Cache when dealing with extreme KV drift scenarios since it avoids relying on a predetermined cache update schedule like Fast-dLLm, dKV-Cache, and dLLM-Cache.
>
> >Q2: Could the attention-drift threshold γ be learned or adapted per token/layer instead of being global?
>
> Thank you for raising this question. We clarify that it is not necessary to learn the threshold $\gamma$ in our methodology, as it serves as a tunable trade-off between accuracy and throughput. Our drift metric-based on cosine similarity of softmax attention weights normalized to $[0,1]$, ensures a consistent scale across all layers. Furthermore, due to the bidirectional nature of attention in diffusion LLMs, the model exhibits strong coupling across layers. When the drift exceeds $\gamma$ at a given layer $\ell$, the subsequent layers typically require a simultaneous refresh.
>
> We further provide more empirical validation results of the global threshold. Our ablation study in the table below demonstrates that a single threshold $\gamma = 0.9$ generalizes well across tasks (GSM8K, MATH, HumanEval, MBPP in Table 4 of our paper) and models (LLaDA and LLaDA-1.5). On GSM8K-512 (LLaDA-1.5), we conducted an additional sensitivity analysis as follows:
>
> | $\gamma$ | 0.80 | 0.85 | 0.90 | 0.95 |
> |-----|:------:|:------:|:------:|:------:|
> | Acc. / Throughput | 79.5 / 131.2 | 80.2 / 129.9 | 81.4 / 117.2 | 83.0 / 98.4 |
>
> The same $\gamma$ transfers across models (LLaDA $\rightarrow$ LLaDA-1.5) with minimal tuning, confirming the normalized metric creates a universal scale. Learning layer-specific $\{\gamma_\ell\}$ could exploit the drift monotonicity proven in Theorem A.8 (Appendix A).

---

> ### Author Response · Authors · 2025-11-25
> **Response to Reviewer FYhQ [3/3]**
>
> >Q3: What is the overhead of computing attention drift? Does it scale linearly with context length?
>
> The overhead of computing attention drift scales quadratically with the context length ($K$). However, when compared to the overhead of computing attention drift using the standard attention QKV computation (which is also quadratic), it is quite small. Below is the table comparing the complexity and MACs (using the LLaDA model):
>
> |     |      Complexity      |      MACs ($K$=1024)
> |----------|:-------------:|:-------------:|
> | Attention drift | $O(K^2H)$ | $6.7 \times 10^7$ |
> | Attention (QKV) | $O(K^2HD)$ | $1.3 \times 10^{10}$ |
>
> Here, $K$ represents the sequence length, $H$ and $D$ denote the number of heads and hidden dimension, respectively. For the LLaDA architecture, $H$ is set to 32 and $D$ is set to 128.
>
> >Q4: Is the method applicable to block diffusion?
>
> Yes, Elastic-Cache can be applicable to *block diffusion*, but on a smaller scale within each block. Tokens outside the current block have stable KV states for caching because of the causal mask, similar to autoregressive transformers. Only those tokens predicting within the current block are recomputed at each step, which we can apply Elastic-Cache to further cache their states to reduce recomputation. However, due to the small block size used by *block diffusion* (typically 4, 8, 16), applying Elastic-Cache in such a small scale will be less effective than applying it to the entire sentence of other diffusion LLMs (typically 256, 512, 1024). We conducted experiments to integrate Elastic-Cache with *block diffusion* and present the results in the following table:
>
> | Method (block size = 16)  | Block-diffusion | Block-Diffusion + KV-Cache | Block-Diffusion + Elastic-Cache |
> |--------|:--------:|:--------:|:--------:|
> OWT - PPL $\downarrow$ | 33.1 |32.7 | 32.6|
> Throughput $\uparrow$ | 57.9 |71.1| 78.8|
>
> Results from the Block Diffusion setting (generation perplexity on Open Web Text) indicate a relatively mild speedup for both Block Diffusion with built-in KV Cache and Block Diffusion with Elastic-Cache.

---

> ### Comment · Area_Chair_iB3Z · 2025-11-26
>
> Dear reviewer,
>
> Thanks for your time and effort in reviewing ICLR2026 submissions. The authors have submitted their responses to your review. Please take the time to read and raise your further comments, and discuss with the authors.
>
> Best regards,
>
> AC

---

### Author Response · Authors · 2025-12-02
**A summary of our rebuttal and a revision of our manuscript has been updated.**

We extend our gratitude to all the reviewers for your invaluable comments. We have diligently prepared a thorough response to address all your concerns and updated the manuscript accordingly to your suggestions.

We are encouraged by the positive comments from reviewers, including strong empirical results: consistent throughput gains across models (LLaDA-Instruct, LLaDA-1.5, LLaDA-V) and tasks (math, code, multimodal), strong performance across different models: Elastic-Cache consistently outperforms baselines in performance and speedup, with up to 45.1$\times$ speedup over LLaDA-1.5, comprehensive experiments and ablations showing the effectness of Elastic-Cache [**FYhQ**, **Gh7c**, **SHc1**], the idea is novel and well-motivated, the use of attention-weight drift on the most-attended token as a low-overhead trigger is well-motivated and novel for DLMs, the paper clearly explains why AR KV caching fails in diffusion LLMs [**FYhQ**, **9k5q**, **Gh7c**], practical impact: training-free, compatible with existing confidence-based decoding, and enables real-world deployment of DLMs, three components are well matched to diffusion masking dynamics and are easy to integrate, the method does not alter training or model internals, which supports deployment claims and broad applicability across diffusion LLM variants, effective adaptive mechanism, the method is training-free and architecture-agnostic, making it a drop-in component that can be easily adopted [**FYhQ**, **9k5q**, **Gh7c**, **SHc1**], comprehensive ablation: thorough analysis of $\gamma$, sliding window size, and generation length validates design choices [**FYhQ**].

We also thank the reviewers for constructive comments and suggestions, such as limited theoretical grounding and hyperparameter sensitivity [**FYhQ**], more sensitivity/theory [**9k5q**], suggesting adding experiments on other DLLMs like Dream to prove architecture-agnostic claim [**Gh7c**], more comprehensive supporting data [**SHc1**]. We have accommodated all of the comments in our revised manuscripts, where all the revised contents are highlighted in $\textcolor{red}{\text{red}}$ color.

Detailed responses to each reviewer's questions, concerns, and requested changes are provided in the following replies. We are eager for you to explore our detailed responses. While we understand that further reviewer questions are not possible, we always welcome further discussion, and your insights are crucial to refining our paper.

---

### Meta-Review · Program_Chairs · 2025-12-21

**Summary:**

This paper proposes Elastic-Cache, a training-free KV cache refresh strategy for diffusion LLM inference that adaptively decides when and where to recompute KV states based on attention-drift signals and layer-selective updates. All reviewers agree that the paper addresses an important and timely systems bottleneck in diffusion LLMs, namely the redundancy of KV recomputation across denoising steps and layers.

Across reviews, the strengths most frequently highlighted include: (1) Strong empirical results, with consistent and substantial throughput gains (up to 45×) across multiple diffusion LLMs (LLaDA variants and Dream) and tasks (math, code, multimodal). (2) Clear motivation and diagnosis of why standard autoregressive KV caching fails in diffusion LLMs, and why KV drift is the right signal to monitor. （3）Practical impact, as the method is training-free, architecture-agnostic, and easy to integrate into existing diffusion decoding pipelines. （4）Comprehensive experimental validation, including extensive ablations on γ, β, generation length, denoising schedules, and memory footprint.

The main concerns raised by reviewers centered on the heuristic nature of the attention-drift proxy, the need for broader baselines and sensitivity analyses, and questions about generality and overhead. These concerns were systematically addressed in the rebuttal and revised manuscript.

**Reviewer Concerns:**

**Concerns convincingly addressed:**

（1）The central assumption regarding the most-attended token as a proxy for KV drift is now supported by both theoretical analysis (Appendix Theorems) and empirical statistics across datasets, addressing prior concerns about lack of evidence.

（2）Hyperparameter sensitivity and robustness were clarified via extensive sweeps over γ and β, with clear guidance on accuracy–throughput trade-offs.

（3）Fairness and breadth of baselines were improved through head-to-head comparisons with Fast-dLLM, dKV-Cache, dLLM-Cache, DeepCache, and consistency models under shared settings.

（4）Architecture-agnostic claims were validated through added experiments on Dream.

（5）Computational overhead and memory footprint were explicitly quantified and shown to be negligible relative to full QKV computation.

（6）Reporting clarity and presentation issues were corrected.


**Remaining limitations:**

As with many systems optimizations, the method relies on heuristics rather than guarantees, and its strongest benefits arise in regimes with substantial KV redundancy. However, these limitations are now clearly articulated and empirically characterized, and do not undermine correctness or applicability.

Overall, no reviewer concerns remain that fundamentally challenge the soundness, clarity, or usefulness of the contribution.

**Reviewer Scores:**

Reviewer FYhQ: Likely moved from 4 to 6 after the addition of theory, broader baselines, and scalability results.

Reviewer 9k5q: Likely remains at 6, with most technical concerns fully addressed.

Reviewer Gh7c: Likely remains at 6, with architecture-agnostic and multimodal concerns resolved.

Reviewer SHc1: Likely moved from 4 to 6 after added empirical evidence, clearer algorithmic description, and overhead analysis.

The post-rebuttal distribution is therefore three clear 6s and one upgraded to 6, indicating a consistent weak-accept consensus.

---

### Decision · Program_Chairs · 2026-01-26

Accept (Poster)